# Neural mechanisms underlying expectation-dependent inhibition of distracting information

Dirk van Moorselaar[1,2,3,4]*, Eline Lampers[1], Elisa Cordesius[1], Heleen A Slagter[1,2,3,4]

[1]Department of Psychology, University of Amsterdam, Amsterdam, Netherlands; [2]Amsterdam Brain and Cognition, University of Amsterdam, Amsterdam, Netherlands; [3]Department of Experimental and Applied Psychology, Vrije Universiteit Amsterdam, Amsterdam, Netherlands; [4]Institute of Brain and Behaviour Amsterdam, Amsterdam, Netherlands

**Abstract** Predictions based on learned statistical regularities in the visual world have been shown to facilitate attention and goal-directed behavior by sharpening the sensory representation of goal-relevant stimuli in advance. Yet, how the brain learns to ignore predictable goal-irrelevant or distracting information is unclear. Here, we used EEG and a visual search task in which the predictability of a distractor's location and/or spatial frequency was manipulated to determine how spatial and feature distractor expectations are neurally implemented and reduce distractor interference. We find that expected distractor features could not only be decoded pre-stimulus, but their representation differed from the representation of that same feature when part of the target. Spatial distractor expectations did not induce changes in preparatory neural activity, but a strongly reduced Pd, an ERP index of inhibition. These results demonstrate that neural effects of statistical learning critically depend on the task relevance and dimension (spatial, feature) of predictions.

*For correspondence:
dirkvanmoorselaar@gmail.com

**Competing interests:** The authors declare that no competing interests exist.

**Reviewing editor:** Joy Geng,

## Introduction

The ability to ignore distracting information is a key component of selective attention, and critical to goal-directed behavior. Recent studies suggest that while attention can be flexibly directed at goal-relevant information, suppression of distracting information is not under similar voluntary top-down control (*Chelazzi et al., 2019*; *Noonan et al., 2018*), as was previously generally assumed (*Jensen and Mazaheri, 2010*). Rather, this work shows that distractor suppression strongly depends on distractor-based learning, for example, about its likely location in the visual environment (*Failing et al., 2019a*; *Ferrante et al., 2018*; *Goschy et al., 2014*; *Reder et al., 2003*; *Sauter et al., 2018*; *Wang and Theeuwes, 2018a*) or its non-spatial features, for example its color or shape (*Cunningham and Egeth, 2016*; *Stilwell et al., 2019*; *Vatterott and Vecera, 2012*). Based on these observations, it has been proposed that distractor inhibition may be dependent on expectations derived from past experience about the likelihood of events (*Noonan et al., 2018*; *Moorselaar and Slagter, 2020*). This idea is grounded in recent notions of predictive processing (*Friston, 2009*) in which the brain continuously uses statistical regularities in the environment to predict the outside world with the overall aim of reducing the mismatch between its a priori predictions and the sensory input. In this framework, processing of any expected stimulus, whether relevant or irrelevant, is reduced (as it elicits a smaller prediction error). Distractor expectations should hence reduce subsequent distractor processing and interference. Yet, it remains unclear how learned inhibition, which

often occurs implicitly (*Ferrante et al., 2018*; *Wang and Theeuwes, 2018a*), is neurally implemented.

Many studies have shown that attention can bias sensory processing of expected task-relevant (or target) information in advance, both on the basis of non-spatial feature (*Battistoni et al., 2017*) and location information (*Giesbrecht et al., 2006*; *Hopfinger et al., 2000*; *Kok et al., 2012*). Yet, it is currently unclear if expectations about upcoming distracting information can similarly prepare the system in advance by modifying activity in sensory regions representing the expected distractor location or its non-spatial features, or alternatively, that distractor learning only becomes apparent upon integration with incoming sensory information (*Moorselaar and Slagter, 2020*). Thus far, this question has only been directly addressed in the context of spatial distractor expectations, with mixed findings. One recent EEG study reported increased pre-stimulus alpha-band activity over visual regions representing the likely distractor location, which was interpreted to reflect top-down preparatory inhibition (*Wang et al., 2019*). In contrast, two other recent EEG studies did not observe any learning-induced changes in preparatory activity of visual regions representing the expected distractor location, as captured by pre-stimulus alpha-band activity (*Noonan et al., 2016*) or by changes in anticipatory spatial tuning as revealed by inverted encoding modeling based on alpha-band activity (*van Moorselaar and Slagter, 2019*). This discrepancy in findings may be explained by the fact that in the former study by *Wang et al., 2019* distractor learning could only be location-based, as the shape and color of the distractor stimulus varied from trial to trial, while in the latter two studies distractor features were fixed, also permitting build-up of feature-based expectations. This may have led to stronger location-based learning in the *Wang et al., 2019* study. Indeed, in this study, it was also found that even targets occurring at the likely distractor location were inhibited, as reflected by slower response times. Behavioral studies have also shown, however, that when targets and distractors are assigned unique and fixed features, the suppression at high-probability distractor locations becomes specific to distractors (*Allenmark et al., 2019*; *Zhang et al., 2019*), indicative of feature-based inhibition. Together, these findings suggest that the locus of distractor learning in the processing hierarchy is flexible and depends on the extent to which expectations are location- and/or feature-specific.

It is currently highly controversial whether expected distractor features can also be represented as so called 'templates for rejection' (*Woodman and Luck, 2007*). Although some behavioral findings are in line with this idea (*Arita et al., 2012*; *Carlisle and Nitka, 2018*; *Park et al., 2007*; *Woodman and Luck, 2007*), to date there is no neural evidence for feature-based distractor templates that set up the system in advance to inhibit distractor processing. However, the few studies that investigated this, cued the distractor on a trial-by-trial basis (*de Vries et al., 2019*; *Reeder et al., 2018*), preventing distractor learning and leaving open the possibility that anticipatory distractor feature templates only arise after repeated encounters with the to-be-ignored feature, that is when statistical learning is possible.

In the current study, we aimed to determine if and how location- and feature-based distractor expectations may independently or in interaction bias sensory regions in advance to suppress distractor processing using visual search tasks and EEG. Across visual searches the distractor appeared more frequently on one location (i.e. high-probability distractor location) than on all other locations (i.e. low-probability distractor locations). Critically, the target and distractor either shared the same spatial frequency, or had different spatial frequencies, that could either be predicted in advance or not (see *Figure 1*). This allowed us to investigate whether preparatory suppression of activity in brain regions representing the high-probability distractor location, as reflected in pre-stimulus alpha-band activity, is dependent on expectations at the feature level. In addition, using multivariate decoding, we could determine if learning about upcoming distractor features (i.e. their spatial frequency) modulated activity of visual regions representing these features in advance, suggestive of a template for rejection.

Although we specifically set out to find anticipatory markers of expectation-based distractor location and feature suppression, it is important to realize that the absence of such effects does not necessarily imply that inhibition is purely reactive. It has recently been proposed that salient distractors can be filtered out pre-attentively (*Gaspelin and Luck, 2018b*; *Sawaki and Luck, 2010*). This so-called signal suppression hypothesis is largely based on the observation that in some contexts, salient distractors can be suppressed, as indicated by a Pd, an ERP marker linked to inhibition, before they capture attention, as indicated by the absence of a distractor-evoked N2pc (*Burra and*

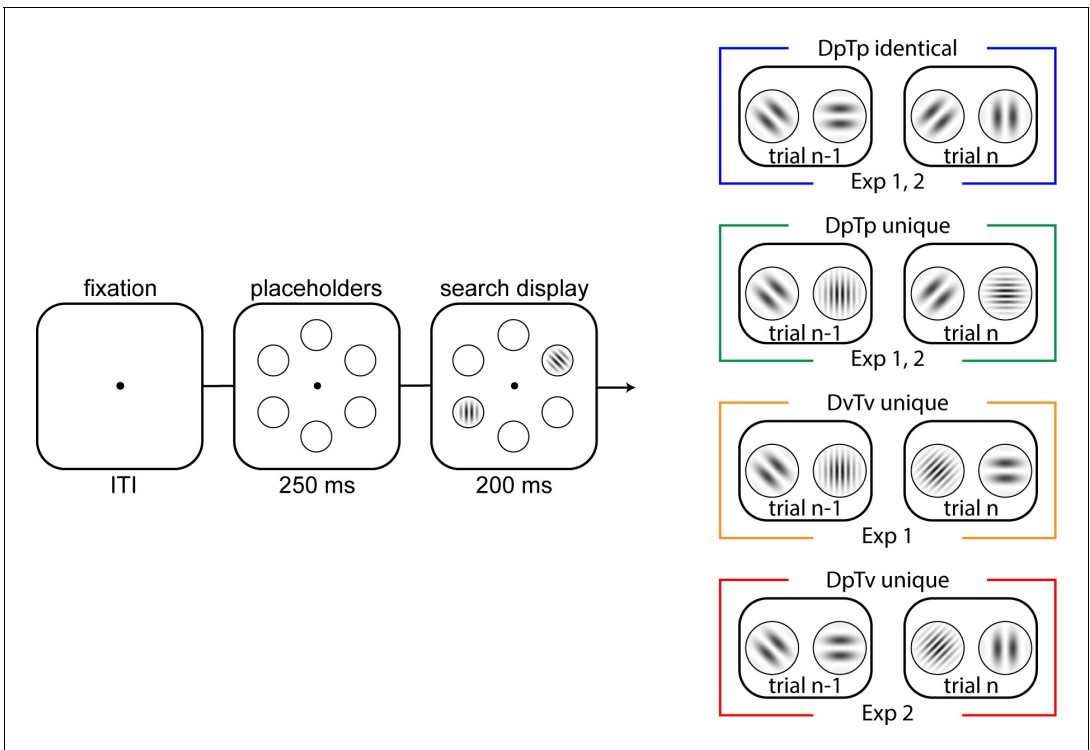

**Figure 1.** Task design of Experiments 1 and 2. Across conditions (colored boxes on the right), in each trial, participants had to indicate the orientation tilt (left or right) of a target gabor. In all conditions, a distractor (a gabor that was horizontally or vertically oriented) could be concurrently presented, which appeared with 70% probability on one specific location. The search display was presented for 200 ms, and participants had 1000 ms to respond. In the DpTp identical and DpTP unique (Experiments 1 and 2) conditions, the spatial frequency of the target and distractor were fixed within an experimental block of trials, with targets and distractors having the same spatial frequency in the former and different spatial frequencies in the latter. In DvTv unique condition (Experiment 1), the target and distractor had different spatial frequencies, which were randomly selected across trials. Finally, in the DpTv condition (Experiment 2), the target and distractor had different spatial frequencies, but only the distractor spatial frequency was fixed across trials in an experimental block. The visualized spatial frequencies do not correspond to the frequencies used in the experiment. Note further that the colors of each condition correspond to condition-specific colors in subsequent figures.

*Kerzel, 2013*; *Gaspar and McDonald, 2014*; *Gaspelin and Luck, 2018a*; *Jannati et al., 2013*; *Wang et al., 2019*). Intriguingly, we recently found that in a context that enforced learning at the spatial level, as distractors and targets were very similar at the feature level (i.e. same spatial frequency), distractors at expected locations continued to elicit an N2pc, but the subsequent Pd was virtually eliminated. We interpreted this as if there was no more need for post-distractor inhibition once participants had learned that information at a specific location could be safely ignored (*van Moorselaar and Slagter, 2019*). Here, we therefore also examined whether a similar pattern is observed when spatial expectations can be combined with feature expectations, or whether distractor-feature specific learning does allow for pre-attentive suppression.

In brief, our results demonstrate that expectations about features of upcoming distractors in the environment induce pre-stimulus sensory templates. Crucially, these sensory templates differed as a function of whether the feature was a predictable feature of the target or of the distractor, providing evidence for distinct target and distractor templates. Yet, we find no evidence for a preparatory spatial bias as a function of distractor location predictability, as no modulations of pre-stimulus alpha-band activity were observed. Distractors at expected locations did evoke a smaller Pd, indicative of a reduced need for inhibition, independent of distractor feature expectations. Thus, neural effects of predictions based on regularities in the environment critically depend not only on the task relevance (target, distractor), but also the dimension (spatial, feature) of these regularities.

## Results

We conducted two experiments, a behavioral experiment and an EEG experiment. The methods and analyses of both these experiments were preregistered at the Open Science Framework (https://osf.io/7ek45/).

## Experiment 1 - Is spatial suppression modulated by feature expectations?

We first conducted a behavioral experiment to examine how distractor expectations at the feature and spatial level may interact in reducing distractor interference. We specifically predicted that suppression at high-probability distractor locations would become specific to distractors (i.e. not affect target processing) when distractors are assigned unique and fixed spatial frequencies, as has been shown previously using shapes and colors (*Allenmark et al., 2019*; *Zhang et al., 2019*). To this end, observers (*N* = 18) performed a visual search task in which the distractor and the target either had the same spatial frequency and only differed in orientation (referred to as Distractor predictable Target predictable [DpTp] identical condition), or differed both in spatial frequency and orientation. In this latter scenario, target and distractor spatial frequencies were either selected at random (referred to as Distractor variable Target variable [DvTv] unique condition) or fixed across a sequence of trials (referred to as DpTp unique condition). This allowed us to test whether learning selectively occurs at the spatial level when search is necessarily feature non-specific, either because target and distractors share the same spatial frequency or because their spatial frequency varies randomly, but not when distractor features can be predicted in advance during feature-specific search (see *Figure 1*).

### Behavior

*Figure 2A* (left plot) shows the mean reaction times (RTs) for the search task as a function of the distractor location for all conditions. These measures were entered into a two-way repeated measures ANOVA (*N* = 18) with the within-subjects factors Condition (DpTp identical, DvTv unique, DpTp unique) and Distractor Location (high-probability distractor location, low-probability distractor location). Replicating previous findings (e.g. *Ferrante et al., 2018*; *Wang and Theeuwes, 2018a*), RTs were faster when the distractor appeared at high relative to low-probability distractor locations (main effect Distractor location; $F_{(1, 17)}=97.0$, p<0.001, $\eta_p^2 = 0.85$). Moreover, there was a main effect of Condition ($F_{(2, 34)}=15.8$, p<0.001, $\eta_p^2 = 0.48$) reflecting faster RTs in the DpTp unique condition, in which spatial frequencies of both the target and distractor could be anticipated, relative to conditions in which those spatial frequencies could not guide search, either because the target and distractor spatial frequency were identical or because they varied from trial to trial. Critically, a significant interaction ($F_{(2, 34)}=5.2$, p=0.011, $\eta_p^2 = 0.23$) showed that the RT benefit at high-probability distractor locations was modulated by the extent to which distractor- and target-specific predictions could be formed at the feature level. Planned comparisons revealed that while distractors were more efficiently ignored on high-probability distractor locations across all conditions (all *t*'s > 6.7, all *p*'s < 0.001, all *d*'s > 1.58), RTs were reduced most by distractor location predictability when distractors and targets could not be dissociated at the feature level in advance. Specifically, in feature specific search (i.e. DpTp unique) spatial benefits were smaller than during feature non-specific searches (DpTp unique vs. DpTp identical, $t_{(17)}=3.0$, p=0.008, $d = 0.71$; DpTp unique vs. DvTv unique, $t_{(17)}=2.0$, p=0.06, $d = 0.48$; $BF_{10} = 1.3$), whereas spatial benefits did not differ between the two feature non-specific search conditions (DpTp identical vs. DvTv unique, $t_{(17)}=1.3$, p=0.22, d = 0.30; $BF_{01} = 2.1$). These findings are indicative of stronger suppression at the high-probability distractor location in the condition in which expectations could necessarily only develop at the spatial level.

To ensure that the observed reductions in distractor interference at high-probability distractor locations do not merely reflect intertrial location priming (*Maljkovic and Nakayama, 1994*), we repeated the preceding analysis after excluding all trials in which the distractor location repeated. This control analysis mimicked the main findings, although the main effects of Condition and Distractor Location (all *F*'s > 16.4, all *p*'s < 0.001, all $\eta_p^2$'s > 0.49) were no longer accompanied by a significant interaction ($F_{(2, 34)}=1.9$, p=0.17, $\eta_p^2 = 0.10$; $BF_{01} = 4.9$). Importantly, however, across all conditions distractors were more efficiently ignored on high-probability distractor locations (all *t*'s > 2.6, all p's < 0.018, all *d*'s > 0.62) demonstrating that intertrial location priming could not

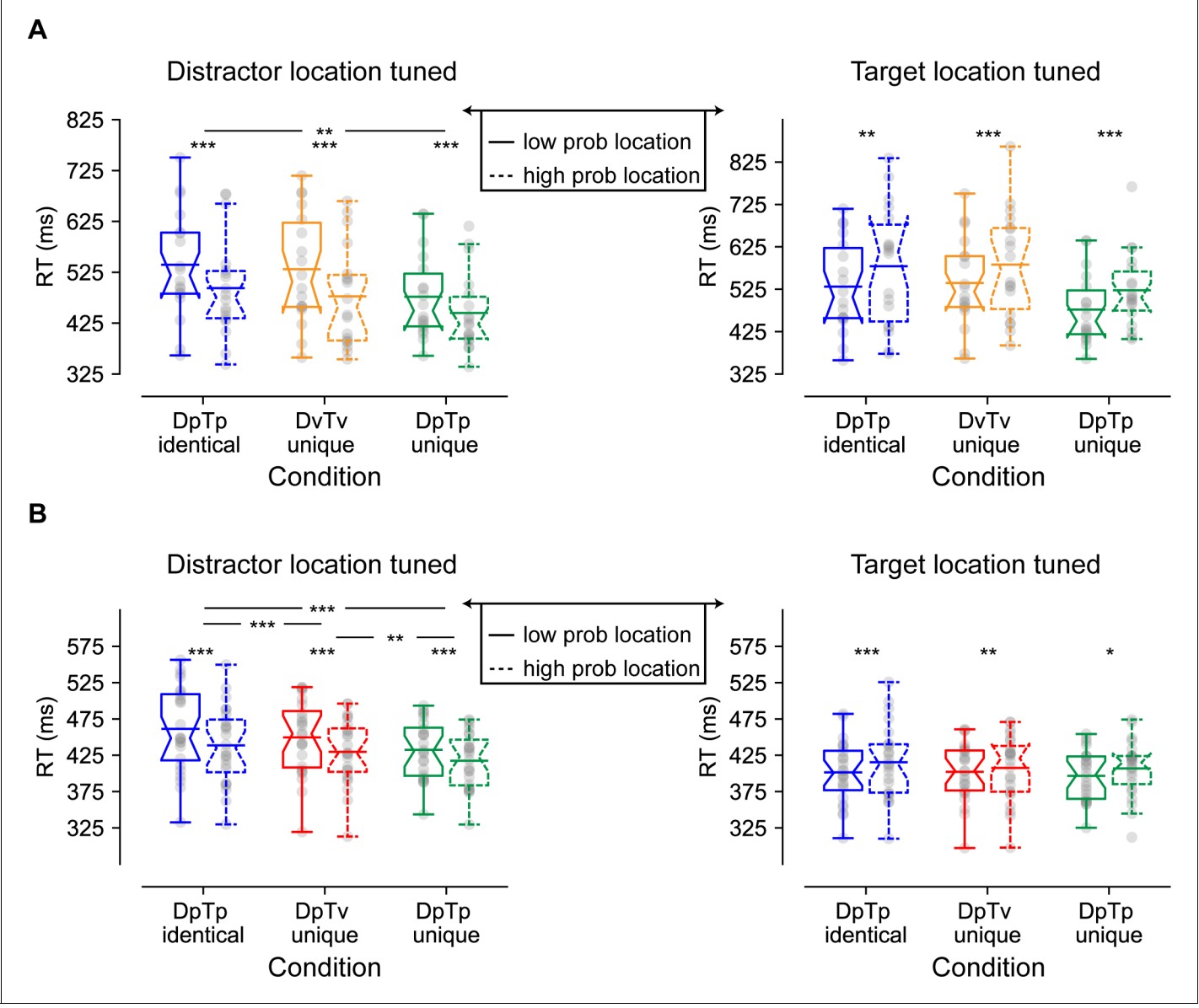

**Figure 2.** Behavioral findings of Experiments 1 (**A**) and 2 (**B**) visualized by notched boxplots with solid horizontal lines corresponding to the mean. (**A**) Reaction times as a function of distractor location (left) and target location (right) in Experiment 1. (**B**) Reaction times as a function of distractor location (left) and target location (right) in Experiment 2, where the target location tuned plot only is based only on distractor absent trials. All distractor location tuned plots do not contain trials with the target at the high-probability location, and the target location tuned plots do not contain distractors at high-probability locations. DpTp indentical: the distractor and the target had the same predictable spatial frequency (fixed across a sequence of trials); DvTv unique: targets and distractors had different, but unpredictable spatial frequencies (varied across a sequence of trials). DpTp unique: targets and distractors had different and predictable spatial frequencies (fixed across a sequence of trials). DpTv unique: targets and distractors had different spatial frequencies, but only the distractor spatial frequency was predictable (fixed across a sequence of trials).

entirely account for the observed reduction in distractor interference at the high-probability distractor location, as also reported in previous studies (e.g. *Failing et al., 2019b*; *Ferrante et al., 2018*).

To examine whether the efficiency of target selection was also affected by distractor location and/or feature foreknowledge, we repeated the same analysis but now tuned to the target location (see *Figure 2A*; right). Consistent with generic suppression at the high-probability distractor location, target processing was also impaired at high-probability distractor locations (main effect Distractor location; $F$ (1, 17)=28.2, p<0.001, $\eta_p^2 = 0.62$). However, this impairment did not differ between conditions (Condition by Distractor Location interaction: $F$ (2, 34)=0.09, p=0.92,

$\eta_p^2 = 0.005$; $BF_{01}$ = 6.7) suggesting that processing of any target was suppressed to the same extent at the high-probability distractor location independent of its similarity to the distractor at the feature level or whether participants had advance knowledge about its spatial frequency.

To summarize, Experiment 1 showed that (1) distractor interference is most reduced when distractors can be predicted at both the spatial and feature level; (1) distractor location learning is more pronounced when distractors can only be predicted on the basis of location, but not feature information; and that (3) any stimulus presented at the expected distractor location is automatically inhibited, including targets. Specifically, target processing at the high-probability distractor location was slowed uniformly across conditions (independent of feature expectations), while the effects of spatial expectations on distractor interference, although present in each condition, were most pronounced in the absence of any feature expectations. These findings suggest that while the level of suppression tied to the high-probability location was fixed, subsequent inhibition of stimuli at the likely distractor location is modulated by expectations about what search items look like at the feature level. It should be noted, however, that the observed interaction between spatial and feature expectations in the distractor-tuned analysis disappeared when we controlled for intertrial spatial priming, an aspect that we will revisit in Experiment 2, which contained many more trials per condition, and hence permitted for the development of more robust spatial and feature expectations.

## Experiment 2: Neural mechanisms underlying learned distractor suppression

The findings from Experiment 1 suggest that the level of the processing hierarchy at which distractor expectations are implemented is dependent on whether expectations can only be developed on the basis of spatial information or can also be grounded in non-spatial regularities. In Experiment 2, we used EEG and a visual search task in which we again manipulated the predictability of a distractor's location and/or spatial frequency to determine how spatial and feature distractor expectations are neurally implemented and may interact in reducing distractor interference. The visual search task was similar to the task used in Experiment 1 with two notable exceptions. In Experiment 1, we found that distractor inference was weakest when both the location and feature of the distractor were predictable. Yet, in this condition (DpTp unique), targets were also predictable at the feature level, rendering it possible that part of this effect is due to target feature expectations. As we were specifically interested in the interaction between distractor learning at the feature and the spatial level and since Experiment 1 showed no difference between the conditions with identical (DpTp identical) and randomly alternating (DvTv unique) spatial frequencies (supposedly both enforcing spatial learning), we replaced the latter condition by a condition in which only the distractor feature was fixed and therefore predictable (DpTv unique). In this condition, the spatial frequency of the target varied from trial to trial and was always different from the fixed and thereby predictable spatial frequency of the distractor. Also, in Experiment 2, we included distractor absent trials, which allowed us to both quantify the magnitude of distractor interference across conditions and get a better estimate of how target processing was affected at predicted distractor locations.

The goal of Experiment 2 ($N$ = 24) was threefold. First, we aimed to establish whether a preparatory alpha gating mechanism contributes to inhibition of the activity of brain regions representing the expected distractor location, and if so, whether such preparatory spatial inhibition is more pronounced in, or even specific to situations in which learning cannot take place at the feature level. Note that using a similar set-up, we recently did not observe any changes in preparatory alpha-band tuning to the expected distractor location (*van Moorselaar and Slagter, 2019*). However, in that study, the distractor's location was repeated across four consecutive trials only, leaving open the possibility that there was simply insufficient time for location predictions to develop. Secondly, we aimed to determine whether preparatory neural activity (also) contains information about the expected feature of the upcoming distractor, and if this is dependent on whether the target is also predictable at the feature level. As noted above, the notion of templates for rejection is currently highly contested. Finally, we aimed to extend our recent finding that distractor location expectations reduced post-distractor inhibition, as indicated by a greatly reduced or even eliminated Pd ERP (*van Moorselaar and Slagter, 2019*), by determining if this effect is also dependent on feature-based distractor expectations or not.

## Behavior

*Figure 2B* shows the mean RTs for the search task as a function of the distractor location for all conditions, which were analyzed with a two-way repeated measures ANOVA with the within-subjects factors Condition (DpTp identical, DpTv unique, DpTp unique) and Distractor Location (high-probability distractor location, low-probability distractor location). As in Experiment 1, distractors at high-probability distractor locations interfered to a lesser extent with target search than distractors at low-probability distractor locations (main effect Distractor Location; $F$ (1, 23)=149.0, p<0.001, $\eta_p^2 = 0.87$), although RTs at high-probability distractor locations were still slower compared to distractor absent trials (all $t$'s > 9.2, all p's < 0.001, all $d$'s > 1.87). As reflected by a main effect of Condition ($F$ (2, 46)=17.5, p<0.001, $\eta_p^2 = 0.43$), overall RTs were fastest when both the target and distractor spatial frequency were predictable and could guide search, intermediate when only the distractor's spatial frequency could guide search and slowest when the spatial frequency of target's and distractors was identical and hence could not guide search. Importantly, and replicating Experiment 1, while distractor interference was reliably reduced at high-probability distractor locations across all conditions (all $t$'s > 10.1, all p's < 0.001, all $d$'s > 2.07), this benefit decreased the more visual search could rely of feature expectations, as captured by a significant Distractor Location by Condition interaction ($F$ (1, 32)=16.5, p<0.001, $\eta_p^2 = 0.42$). Indeed, the benefit of distractor location predictability (i.e. high vs. low) was most pronounced when distractors and targets consistently shared the same spatial frequency and learning could only be location-based ($M = 50.9 \pm$ SD=24.4; DpTp identical vs. DpTp unique: $t$ (23)=4.6, p<0.001, $d = 0.93$; DpTp identical vs. DpTv unique; $t$ (23)=4.1, p<0.001, $d = 0.84$), intermediate when feature prediction could be formulated only for the distractor ($M = 41.7 \pm$ SD=20.1) and smallest when feature predictions could be formulated for both the distractor and target ($M = 30.6 \pm$ SD=10.7; DpTv unique vs. DpTp unique: $t$ (23)=3.0, p=0.005, $d = 0.62$). These results suggest that distractor feature learning may shift the locus in the visual hierarchy at which expectation-based suppression operates from space-based to feature-based.

Next, we again examined whether the observed reductions in distractor interference at high-probability distractor locations were the consequence of statistical learning rather than simply of short-lasting, inter-trial location priming (*Maljkovic and Nakayama, 1994*). After exclusion of all trials in which the distractor location repeated between trials, we again observed main effects of Distractor Location and Condition (all $F$'s > 19.7, all p's < 0.001, all $\eta_p^{2'}s > 0.46$) and, counter to Experiment 1, the interaction also remained reliable (F (1, 32)=9.6, p<0.001,, $\eta_p^2 = 0.29$), indicating that neither the effect of distractor location predictability on visual search times, nor the differences between conditions therein could be simply explained by intertrial spatial priming effects.

Having established that distractors were generally more efficiently ignored at high-probability distractor locations, we next investigated whether as in Experiment 1, target processing was impaired at high-probability distractor locations and if this was dependent on the extent to which targets and/or distractors were predictable at the feature level. For this purpose, we repeated the preceding analyses but only included distractor absent trials. As visualized in *Figure 2B* (right) and consistent with Experiment 1, target detection was hampered at high-probability distractor locations across conditions (main effect Target Location; $F$ (1, 23)=25.4, p<0.001, $\eta_p^2 = 0.53$). As in Experiment 1, this effect was not accompanied by an interaction with Condition suggesting that target processing was impaired to the same extent at high-probability distractor locations irrespective of the nature of feature expectations ($F$ (2, 46)=1.8, p=0.18, $\eta_p^2 = 0.071; BF = 0.07$).

Together, these results replicate the finding of Experiment one that the visual system continues to be sensitive to feature information at a location that is suppressed in advance (*Stilwell et al., 2019*). Whereas target processing was impaired to the same extent across conditions, arguing for a fixed level of suppression tied to the high-probability location, subsequent inhibition of distracting stimuli at this location was modulated by expectations at the feature level, with suppression being most pronounced when learning could necessarily only be space-based. In Experiment 2, this interaction between spatial and feature expectations remained significant even after controlling for spatial intertrial transitions, suggesting that when expectations can be grounded in longer learning, the system combines spatial and feature information about probable upcoming distractors to reduce distractor interference. This makes sense from an ecological perspective: one would not want to simply inhibit anything occurring at a particular location in space.

## EEG

### Time-frequency analyses: pre-stimulus alpha-band activity does not increase contralateral to the high-probability distractor location

At the behavioral level, we found that spatial distractor expectations generally reduced distractor interference during visual search, albeit most strongly so in the absence of distractor feature expectations, and furthermore, that target search was slowed down when the target occurred on the high-probably distractor location, indicative of generic spatial suppression. Using time-frequency analyses, we first examined if location-based distractor expectations modulated activity of visual regions representing the likely distractor location in advance, as indexed by a pre-stimulus alpha-band lateralization index at posterior electrode sides (POS), where contralateral and ipsilateral were relative to the high-probability distractor location:

$$\frac{a_{contralateral\ POS} - a_{ipsilateral\ POS}}{a_{contralateral\ POS} + a_{ipsilateral\ POS}}$$

and if so, whether pre-stimulus alpha-band lateralization reduced when learning could also occur at the feature level. As we were specifically interested in lateralized alpha power as a measure of anticipatory spatial suppression, we limited the statistical analyses to the anticipatory time window (i.e. $-750$ ms $-$ 0 ms). Consequently, the analysis was only sensitive to learned expectations and hence could not be confounded by the actual stimulus configuration, which allowed us to include all trials. The lateralization index, which was analyzed at electrodes PO7/8 and O1/2, has the benefit of not having to select a pre-stimulus baseline window, the same time window where we expected alpha-band modulations. Note, that for the same reason, *Wang et al., 2019* developed a Z-transformation procedure, which we (as preregistered) also applied here. This resulted in virtually identical results as obtained with the lateralization index.

As visualized in *Figure 3A/B*, cluster-based permutation tests across time and frequency and across time for lateralization index averaged within the alpha-band (8–12 Hz) did not show any reliable increase of alpha-power contralateral vs. ipsilateral to the high-probability distractor location in any condition. Nevertheless, *Figure 3B* also suggests that in the DpTp identical condition - the one condition in which learning could only take place at the location level, but not at the feature level - a contralateral increase in alpha-band power was present numerically, albeit not significantly so. To increase sensitivity to a potentially weak effect, in an exploratory analysis, we entered the $a$ lateralization index averaged over the entire anticipatory time window into a repeated measures ANOVA. Critically, this analysis did not show a main effect of Condition ($F$ (2, 46)=0.4, p=0.68, $\eta_p^2 = 0.016$; $BF_{01}$ = 5.70), neither did the $a$ lateralization index differ from zero in the DpTp identical condition ($t$ (23)=1.2, p=0.21, $d$ = 0.27; $BF_{01}$ = 2.22). Thus, these time-frequency findings suggest that the observed spatial suppression at the high-probability distractor location at the behavioral level was not mediated by a contralateral increase of alpha-band power in anticipation of search display onset, even in a condition in which learning could only occur at the spatial level and across a longer time span than in two previous studies that also did not observe distractor learning-related changes in pre-stimulus alpha-band activity (*Noonan et al., 2016*; *van Moorselaar and Slagter, 2019*), but in contrast to another study (*Wang et al., 2019*).

### Decoding: target and distractor-based feature expectations induce pre-stimulus sensory templates

Our behavioral findings showed that distractor feature-specific expectations in the absence of any target expectations modulated spatial expectation-dependent suppression, suggesting that non-spatial information about upcoming distractors reduced distractor interference. To investigate whether feature-specific distractor expectations induced pre-stimulus sensory templates, that is were already evident in the anticipatory EEG signal, we trained a classifier on the response pattern of 64 electrodes using each spatial frequency (i.e. low, medium, high) for each timepoint, separately for targets and distractors (see Materials and methods section for details). ERP studies have demonstrated differences in early visual-evoked responses as a function of stimulus spatial frequency (e. g. *Kenemans et al., 1993*; *Proverbio et al., 1996*), an effect that is attributed to macro-scale differences in sensitivity to spatial frequency across visual regions (*Kenemans et al., 2000*). Therefore, one would expect that multivariate pattern analysis techniques are also sensitive to spatial frequency

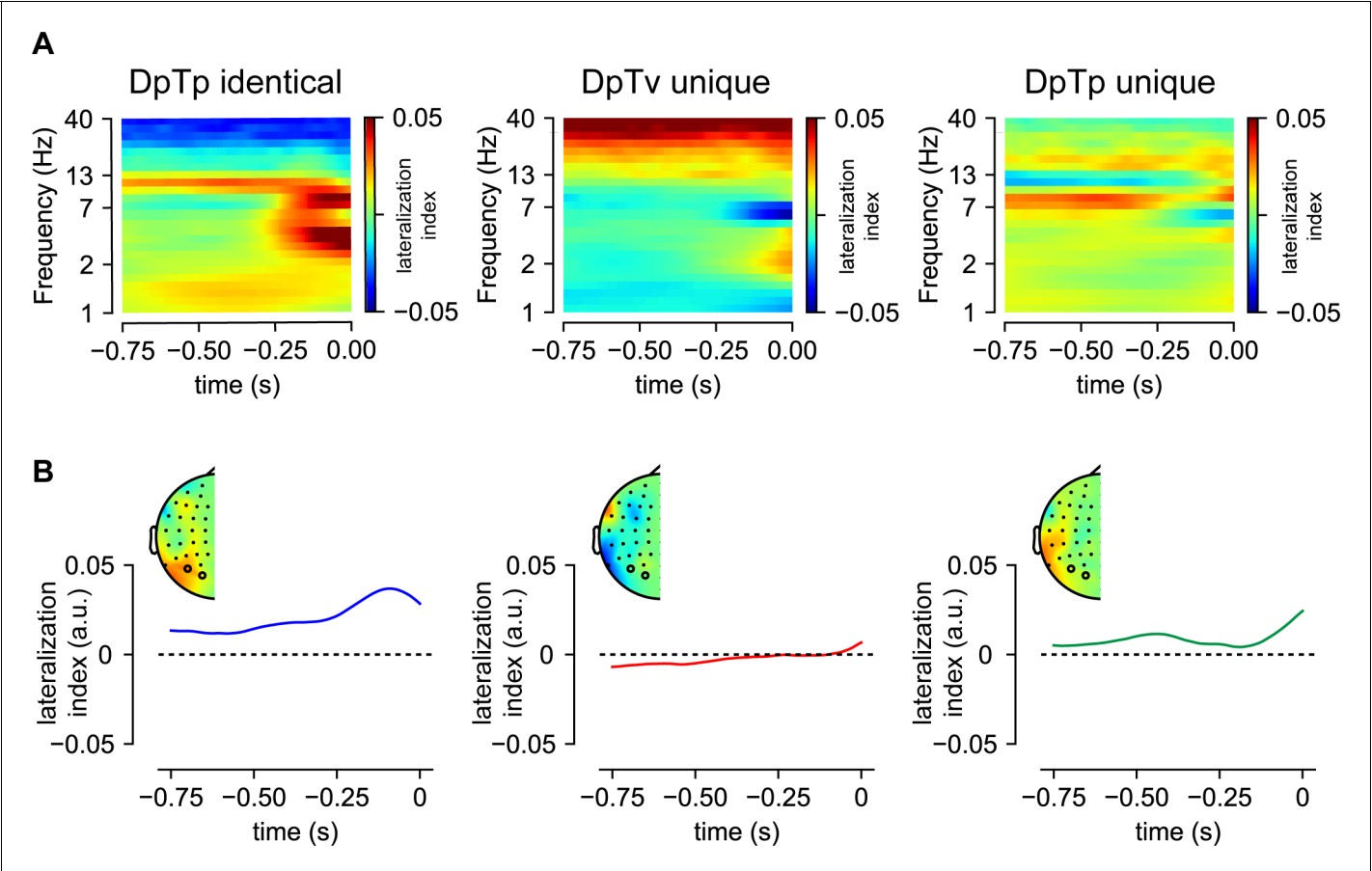

**Figure 3.** Anticipatory alpha-activity. (**A**) These panels show the lateralization index relative to the high-probability distractor location averaged over PO7/8 and O1/2 separately per condition. (**B**) Time-series of lateralization index averaged within the alpha-band (8–12 Hz). Insets show topographic distribution averaged over the entire anticipatory window. Although numerically, pre-stimulus alpha-band activity was higher over visual regions contralateral to the predicted distractor location in the DpTp indentical condition in which the distractor and the target had the same fixed spatial frequency, permitting only the development of spatial distractor expectations, this effect was not statistically significant. DpTv unique: targets and distractors had different spatial frequencies, but only the distractor spatial frequency was predictable (fixed across a sequence of trials); DpTp unique: targets and distractors had different and predictable spatial frequencies (fixed across a sequence of trials).

as measured with EEG. Yet, to our knowledge, no study has so far shown that it is also possible to decode spatial frequency from the pattern of scalp-EEG activity.

Classification performance visualized in *Figure 4* shows that the classifier could discriminate not only the target, but also the distractor's spatial frequency, in all conditions(as it is unclear whether in the DpTp identical condition, in which targets and distractors were of the same spatial frequency, classification was driven by the spatial frequency of the target, of the distractor or both, we did not contrast this condition with the unique feature conditions). Interestingly, both in anticipation and during stimulus processing, target decoding was numerically higher when its spatial frequency was predictable (i.e. DpTp unique condition) compared to when it was not (i.e. DpTv unique condition), whereas the opposite pattern was observed for distractor decoding: the spatial frequency of the distractor could be decoded with higher accuracy when only the spatial frequency of the distractor, but not the target spatial frequency could be predicted in advance (in the DpTv unique compared to the DpTp unique condition). The observation that the anticipatory EEG signal already appeared to contain information about the upcoming distractor is especially interesting as it may support the notion of a negative sensory template or preparatory feature-specific distractor inhibition. Critically, anticipatory target decoding was absent in the DpTv unique condition, the one condition where target expectations could not develop, arguing against the possibility that the observed anticipatory decoding can be explained by an artificial shift of the stimulus-evoked response (***van Driel et al.,***

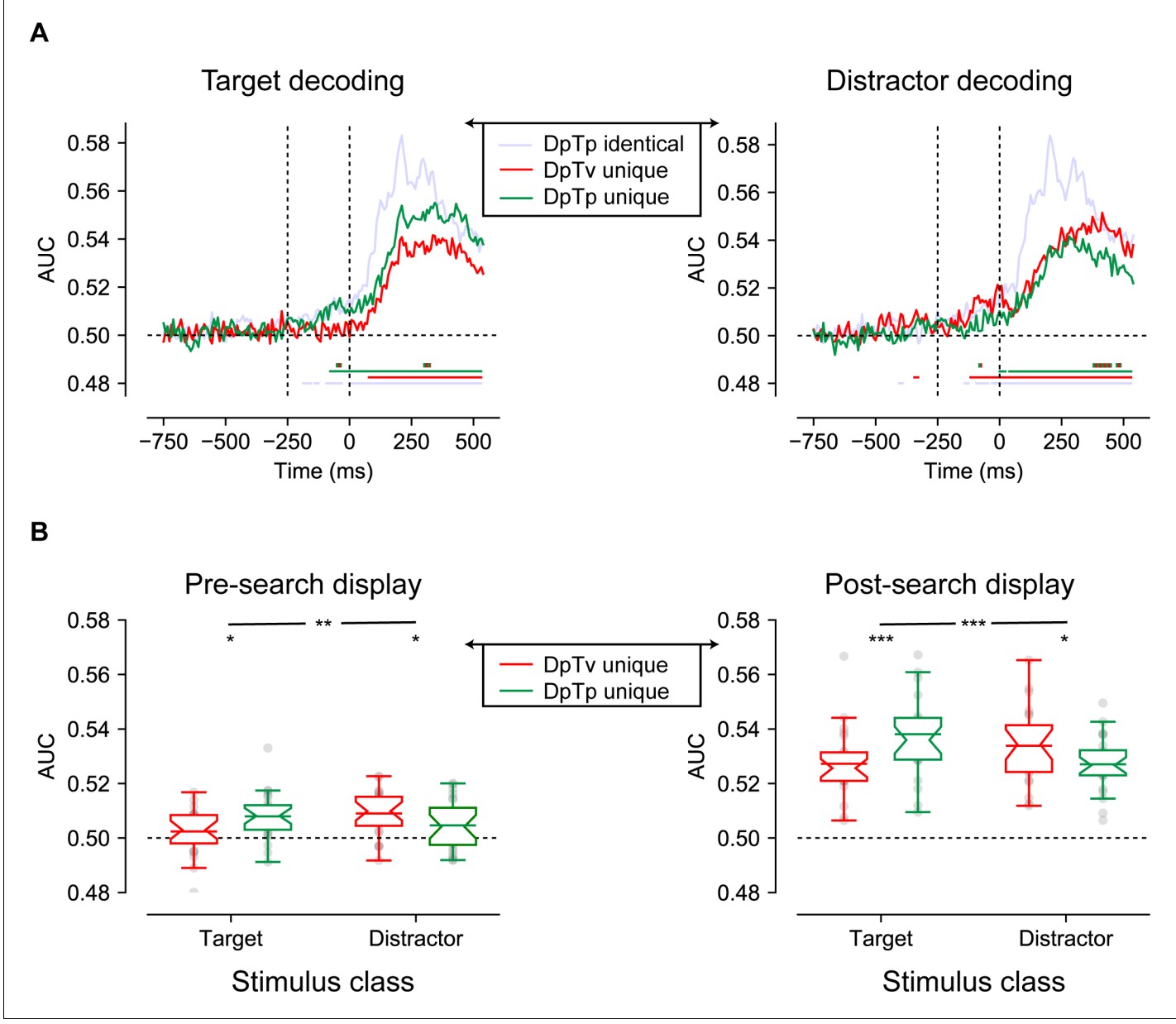

**Figure 4.** Multivariate decoding of the target and distractor spatial frequency across conditions using broad-band EEG of all 64 electrodes. (A) AUC scores of target (left) and distractor (right) decoding across time. Colored bars on the x-axis (blue; red; green) indicate clusters where conditions differ significantly from chance after cluster correction (p<0.05). Red and green dashed lines indicate clusters with a significant difference between DpTv unique and DpTp unique conditions after cluster correction (p<0.05). (B) Averaged auc scores during the pre_search display (−250–0 ms; left) and the post-search display (0–550 ms; right) visualized by notched boxplots with solid horizontal lines corresponding to the mean. Horizontal black dashed lines across plots indicates chance performance, whereas the vertical dashed black lines in the upper plots indicate the onset of placeholder (−250 ms) and the search display (0 ms).

*2019*). Also note that anticipatory above chance decoding was not sustained throughout the entire pre-stimulus period, but only emerged close to search display onset, raising the possibility suggesting that sustained anticipatory expectation signals were subtracted out by the baselining procedure (see Materials and methods). Control analyses, however, using either a condition-specific pre-stimulus baseline (collapsed across spatial frequency) or no baseline at all, yielded identical results indicating that the observed feature specificity in the EEG signal only emerged in close temporal proximity to search display onset.

Cluster-based permutation tests across time confirmed this pattern of observations. As we were specifically interested in anticipatory markers of distractor feature expectations, these tests were performed separately for the anticipatory (−750 to 0 ms) and the reactive (0–550 ms) time windows. As visualized by the red and green colored horizontal lines in *Figure 4A* (p<0.05, cluster-corrected), anticipatory target decoding was specific to the DpTp unique condition, whereas anticipatory distractor decoding was specific to the DpTv unique condition. These condition differences were confirmed by a direct contrast between DpTp and DpTv unique conditions, which yielded anticipatory and reactive differences between conditions, both for target and distractor decoding (p<0.05, cluster-corrected; double-colored lines in *Figure 4A*). These statistical results reveal that when both the targets and the distractors spatial frequency could be predicted, participants especially relied on target feature-specific information to guide search. However, when they could not know what the target would look like, as it randomly varied across trials, and the target should thus also not be evident in the anticipatory neural code, participants relied more on distractor feature-specific information to bias search, and importantly, already in anticipation of search display onset.

To more directly characterize these effects, we followed up with an exploratory analysis in which average decoding performance in separate time windows (pre-search display: −250 ms to 0 ms; or post-search display: 0 ms to 550 ms) was used as the dependent variable in a repeated measures ANOVA with within-subject factors Stimulus Class (target, distractor) and Condition (DpTp unique, DpTv unique). As visualized in *Figure 4B*, and consistent with the pattern described thus far, this revealed significant effects of being able to predict the target and distractor spatial frequency in advance, that differed as a function of condition, both before the onset of the search display (in the preparatory period) and after search display onset, as revealed by significant interactions between Stimulus Class and Condition ($F$ (1, 23)=11.3, p=0.003, $\eta_p^2 = 0.33$ and $F$ (1, 23)=59.3, p<0.001, $\eta_p^2 = 0.72$, respectively). These interactions reflected better target decoding in the DpTp unique than in the DpTv unique condition both before and after target presentation ($t$ (23)=2.1, p=0.044, $d = 0.44$ and $t$ (23)=4.0, p<0.001, $d = 0.82$, respectively) and vice versa for distractor decoding ($t$ (23)=2.1, p=0.046, $d = 0.43$ and $t$ (23)=2.5, p=0.02, $d = 0.51$, respectively). Together, these findings show that we could not only decode the spatial frequency of the target from the pattern of pre-search display brain activity, but also, when the targets spatial frequency was unpredictable, the spatial frequency of the distractor, suggestive of a distractor feature template.

## Cross-class decoding: distractor and target expectations induce distinct pre-stimulus sensory templates

The observed anticipatory distractor decoding is in line with the idea that distractor-specific expectations may help to filter out distractors preemptively. Yet, it leaves open the question how they may do so, as decoding by itself is uninformative about the nature of the underlying neural representation (*Moorselaar and Slagter, 2020*). A hallmark of an advance rejection template is that processing of the coded features is inhibited and the neural representation of a rejection template should thus be distinct from a target-defined attentional template which in contrast facilitates processing of matching stimuli.

To test this prediction, we explored whether the pattern driving anticipatory distractor decoding differed from that driving target decoding. Specifically, we trained the classifier separately on either target or distractor spatial frequencies using the same trials that were used in the preceding analyses. However, we now used the resulting weights to classify the spatial frequency of the target in an independent set of trials, namely target-only trials without a distractor. The logic here is that if targets and distractors are represented via the same neural code, it should not matter which of the two is used in the training stage. Yet, distractor weights should not generalize to targets when they are represented via distinct neural codes.

As visualized in *Figure 5A* (solid lines) and as one would expect, the target trained model tested on target-only trials replicated the findings of the main analysis: the spatial frequency of the target could be reliably decoded post search display in all conditions, but also in anticipation of the search display selectively in the condition in which the spatial frequency of the target was predictable (DpTp unique; right plot). Cluster-based permutation tests across time confirmed that target-target decoding was not only reliable in both the pre- and post-stimulus window (p<0.05, cluster-corrected; solid green horizontal lines), but also differed from distractor-target decoding (p<0.05, cluster-

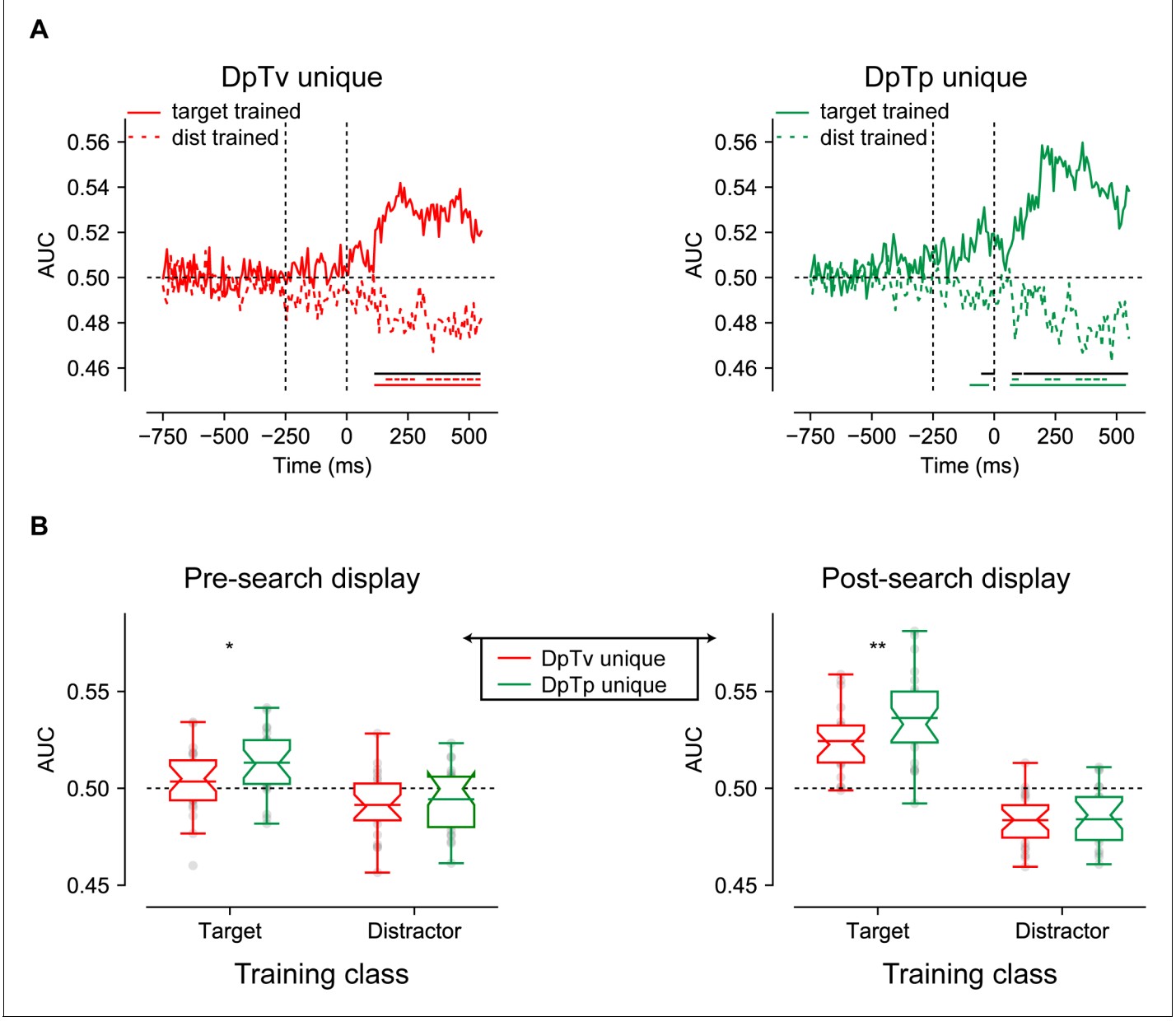

**Figure 5.** Multivariate decoding of the target using either weights from a model trained on targets spatial frequencies and distractor spatial frequencies. (A) AUC scores of DpTv unique (left) and DpTp unique (right) decoding across time. Colored bars on the x-axis indicate clusters where conditions differ significantly from chance after cluster correction (p<0.05). Solid black lines indicate clusters with a significant difference between models trained on targets and distractors after cluster correction (p<0.05). (B) Averaged auc scores during the pre_search display (−250 to 0 ms; left) and the post-search display (0–550 ms; right) visualized by notched boxplots with solid horizontal lines corresponding to the mean. Horizontal black dashed lines across plots indicates chance performance, whereas the vertical dashed black lines in the upper plots indicate the onset of placeholder (−250 ms) and the search display (0 ms).

corrected; black horizontal lines). Indeed, using distractor trained weights, the exact same spatial frequency could no longer be decoded above change in advance when it was a predictable feature of the target, potentially indicative of a distinct distractor template (*Figure 5A*; dashed lines). That is, the predictable target spatial frequency could only be decoded from the pattern of pre-stimulus EEG activity using target-based classification weights, but not using distractor-based classification weights, even though this same spatial frequency could be reconstructed using distractor-based decoding (*Figure 4*). What's more, pre-stimulus target decoding performance of the distractor-trained model did not differ between the conditions in which the target spatial frequency could be

predicted in advance vs. could not be predicted in advance, neither when tested with cluster correction nor with paired t-tests within the pre-search display window ($t$ (23)=0.57, p=0.57, $d$ = 0.12; $BF_{01}$ = 4.03) or the post-search display window ($t$ (23)=0.14, p=0.89, $d$ = 0.028; $BF_{01}$ = 4.61). Nonetheless, the Stimulus Class (target trained, distractor trained) by Condition interaction was not significant in the pre-search display window ($F$ (1, 23)=0.6, p=0.44, $\eta_p^2 = 0.026$; $BF_{01}$ = 2.29; *Figure 5B*). Next to these anticipatory effects, cluster-based permutation tests identified significant below chance decoding during visual search using the distractor trained weights (p<0.05, cluster-corrected; dashed green horizontal lines). Together, these findings suggest that spatial frequency was differently represented in the pattern of scalp EEG activity as a function of whether it was a feature of the target or of the distractor.

## ERPs: the distractor Pd reduces as a function of spatial expectations

Finally, we examined how distractor feature and locations expectations modulated stimulus processing as indexed by the lateralized N2pc and Pd components, signaling attentional selection (*Eimer, 2014*; *Luck, 2012*; *Luck and Hillyard, 1994*) and suppression (*Gaspelin and Luck, 2018a*; *Hickey et al., 2009*), respectively. To isolate distractor- and target-specific components, separate difference waveforms (contralateral – ipsilateral) were computed for lateralized distractors below the horizontal midline with concurrent targets on the vertical midline, or vice versa, using O1/O2, PO3/PO4 and PO7/PO8 as electrodes of interest (see Materials and methods section for details).

As visualized in *Figure 6*, we observed a clear distractor-evoked N2pc in all conditions, which only seemed to be somewhat reduced in amplitude at the high-probability distractor location in the DpTp unique condition. Likewise, as shown in *Figure 7*, the target-evoked N2pc also appeared to be virtually unaffected by feature and/or location expectations. Yet, notably, and in line with our previous study (*van Moorselaar and Slagter, 2019*), the distractor-evoked Pd was virtually eliminated when distractors appeared at high-probability distractor locations, signaling a reduced need for post-distractor inhibition.

To statistically test these effects, voltages within the N2pc window (114–194 ms; see Materials and methods section for details) were submitted to a Stimulus Class (target, distractor) by Condition (DpTp identical, DpTp unique, DpTv unique) by Location (high distractor probability, low distractor probability) by Hemifield (contralateral, ipsilateral) repeated measures ANOVA. This analysis confirmed that the N2pc was reliably evoked across conditions (main effect Hemifield: $F$ (1, 23) =65.5, p<0.001, $\eta_p^2 = 0.74$) and also showed that as could be expected, the N2pc was more pronounced in response to targets than distractors (interaction Stimulus Class by Hemifield: $F$ (1, 23) =37.3, p<0.001, $\eta_p^2 = 0.62$). Critically, in the DpTp unique condition, when participants had foreknowledge about both the targets and distractors features, the distractor N2pc was reduced and the target N2pc increased relative to the other conditions without unique target feature predictions (interaction Stimulus Class by Condition by Hemifield: $F$ (2, 46)=5.3, p=0.009, $\eta_p^2 = 0.19$), indicating that expectations at the target level made attentional selection more efficient. However, there was no evidence whatsoever that location expectations modulated the amplitude of the N2pc, neither for distractors or for targets (all $F$'s < 1.1, all $p$'s > 0.33, all $\eta_p^2$'s > 0.048).

The N2pc results suggest that distractors continued to capture attention to the same extent at high as at low-probability distractor locations, albeit generally not as strongly as targets. Yet, at high-probability locations there appeared to be a reduced need to actively counteract this shift of attention as evidenced by a reduction in the amplitude of the distractor-evoked Pd. To statistically test this, average voltage values within the Pd window (243–323 ms; see Materials and methods section for details) elicited by distractors were submitted to a Condition (DpTp identical, DpTp unique, DpTv unique) by Location (high distractor probability, low distractor probability) by Hemifield (contralateral, ipsilateral) repeated measures ANOVA. Critically, this analysis yielded a Location by Hemifield interaction ($F$ (1, 23)=5.1, p=0.033, $\eta_p^2 = 0.18$), reflecting a reduction in Pd amplitude at high- vs. low-probability distractor locations across conditions. Together these results replicate our previous findings that in a scenario where location expectations do not influence the initial selection of distracting information, they can nevertheless reduce the subsequent need to actively suppress the expected distracting information (*van Moorselaar and Slagter, 2019*). The main effect of Hemifield

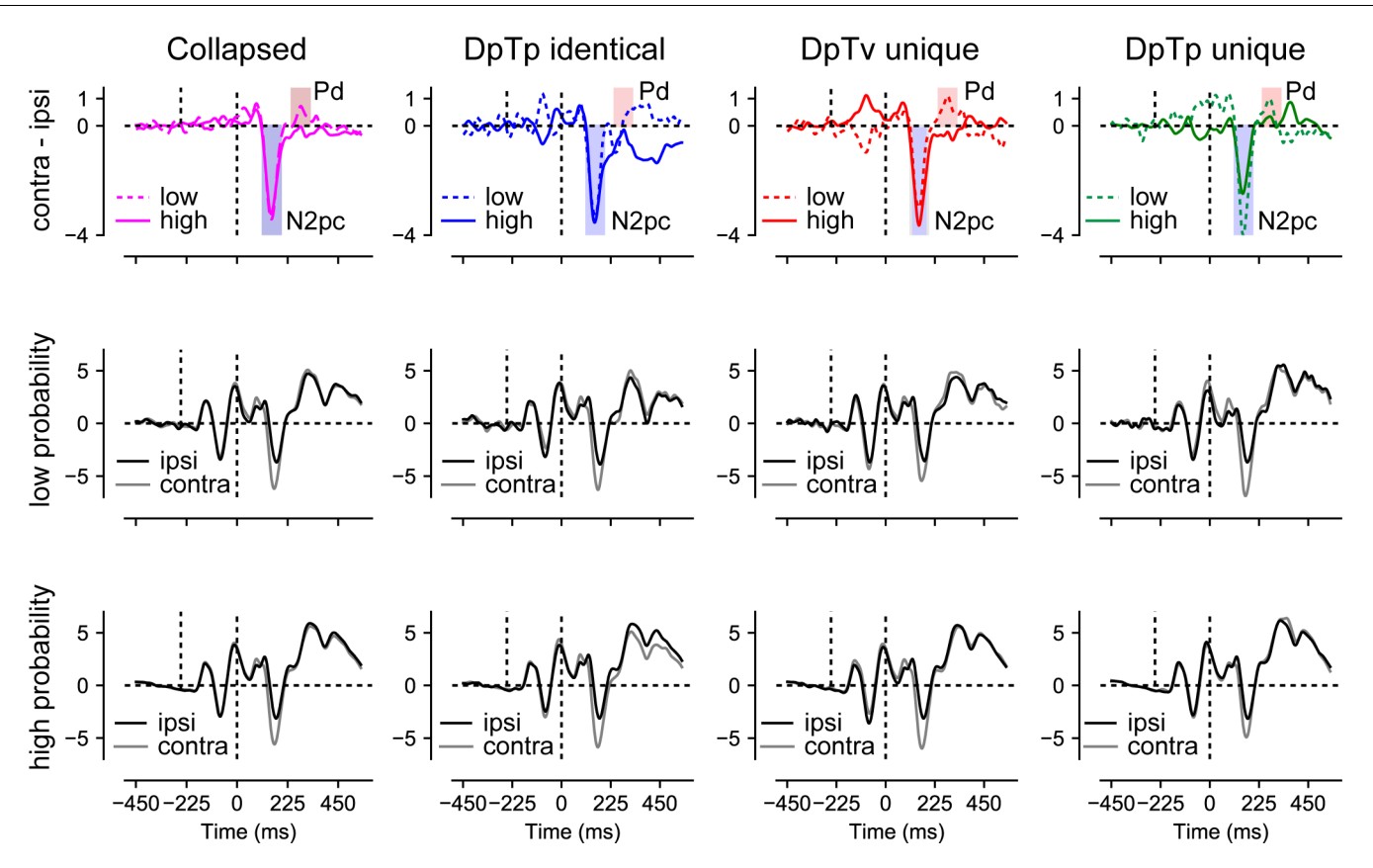

**Figure 6.** Distractor Pd reduces as a function of distractor location expectations. ERP results from search trials with lateral distractors collapsed across all conditions (column 1) and separately for each condition (columns 2–4). Row one shows the difference between contra- and ipsilateral waveforms for the low and high-probability distractor location. Only the distractor-evoked Pd, not the N2pc, was reduced at high- vs. low-probability distractor locations (top left figure). Rows 2 and 3 show the contra- and ipsilateral waveforms for the low- and high-probability distractor locations, respectively. Microvolts are plotted on the y-axes. Blue and red transparent rectangles in row one show the mean averaging windows for N2pc and Pd analyses.

and all other interaction effects with Location were non-significant (all $F$'s < 1.9, all p's > 0.16, all $\eta_p^2$'s > 0.077).

## Discussion

The aim of the present study was to gain more insight into the neural mechanisms underlying learning to ignore distracting information on the basis of spatial and/or non-spatial regularities. While there is evidence that expectations based on regularities in the environment induce pre-stimulus sensory templates (*Kok et al., 2017*; *van Moorselaar and Slagter, 2019*), it remains unclear whether the generation of such sensory templates is dependent on the task relevance of the sensory information, and hence is also evident when distractor rather than target information can be predicted in advance. Here we manipulated the predictability of the distractor at both the spatial and feature level, to address this outstanding question. At the behavioral level, responses were faster when the distractor occurred at the high-probability distractor location, but slower when the target occurred at this location, relative to any other location, indicative of generic expectation-dependent spatial suppression. Moreover, when distractors were also predictable at the feature level, location-based distractor suppression became less pronounced, suggesting that the locus of inhibition in the processing hierarchy is flexible and dependent on whether expectations are location- and/or feature-specific. Yet, at the neural level, even when distractors were not predictable at the feature level, we observed no evidence for preparatory spatial inhibition as a function of distractor location learning, as indicated by the absence of a modulation of pre-stimulus lateralized alpha-band activity. In

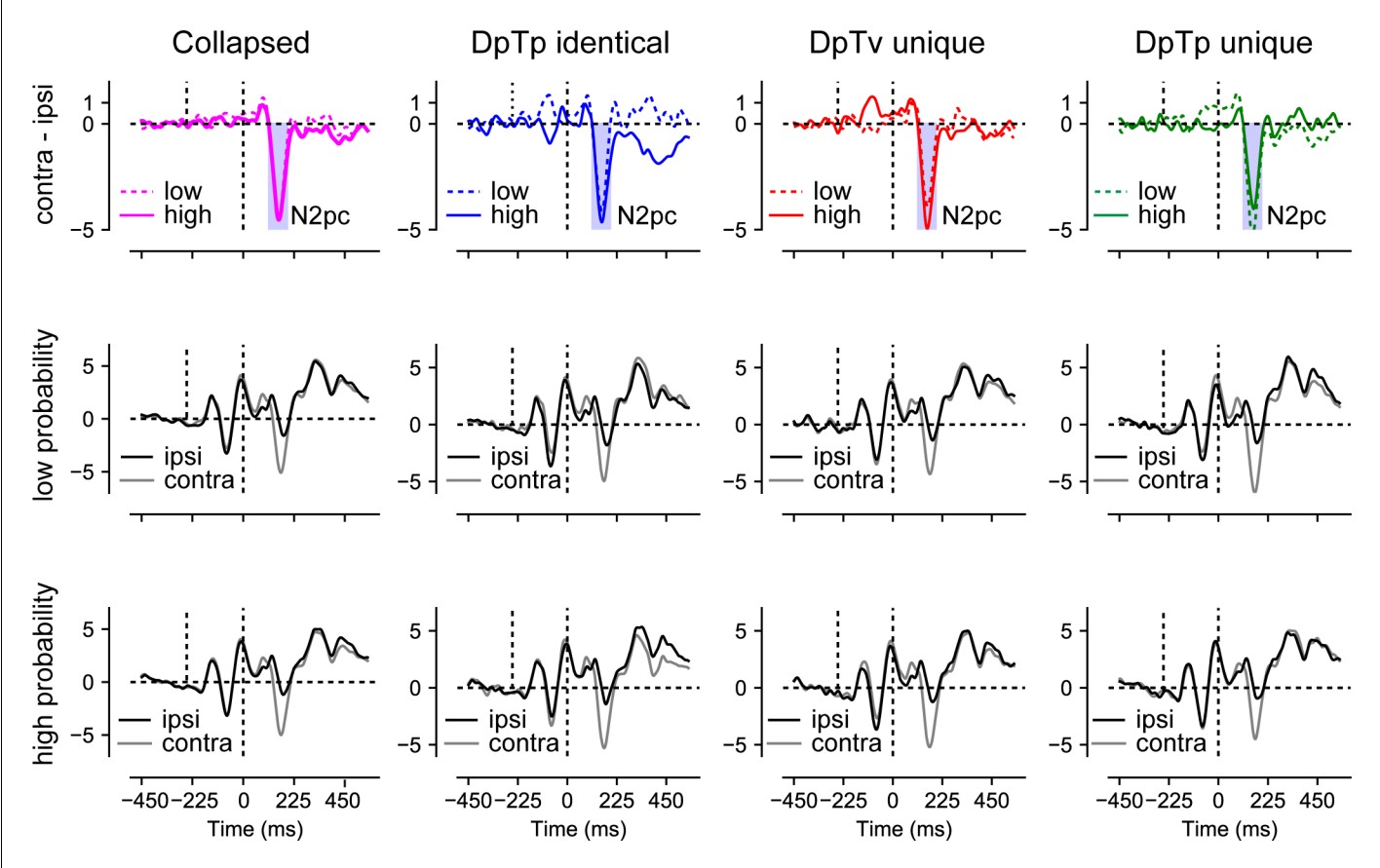

**Figure 7.** No effect of distractor or target expectations on the target-evoked N2pc. This figure displays ERP results from search trials with lateral targets collapsed across all conditions (column 1) and separately for each condition (columns 2–4) and the low- and high-probability distractor locations (rows 2 and 3, respectively). Row one shows the difference between contra- and ipsilateral waveforms for low- and high-probability location. Microvolts are plotted on the y-axes. Blue transparent rectangles in row one show the mean averaging windows for N2pc analyses.

contrast, when distractor features (their spatial frequency) were predictable, they could already be decoded from the pattern of scalp EEG activity before search display onset, potentially indicative of a distractor feature template. Strikingly, spatial frequency was differently represented in the pattern of pre-stimulus scalp EEG activity as a function of whether it was a predictable feature of the target or of the distractor, suggesting distinct target and distractor templates. Spatial distractor expectations did reduce post-distractor inhibition, as indicated by a strongly reduced distractor-evoked Pd. Together these findings demonstrate that location and feature-based expectations in part rely on different neural mechanisms, but also interact, in inhibiting distracting information.

At the spatial level, there is some evidence that pre-stimulus alpha-band activity contributes to expectation-dependent suppression (*Wang et al., 2019*), but based on other work (*Noonan et al., 2016*; *van Moorselaar and Slagter, 2019*), we reasoned that this may be specific to conditions that enforce learning at the spatial level. Counter to this idea, however, here lateralized alpha-band activity was insensitive to distractor location learning, even when the target and distractor shared the same spatial frequency and learning could only be location based. The absence of pre-stimulus alpha-band modulations in any condition despite clear behavioral evidence for generic space-based suppression adds to a growing body of literature challenging the widely embraced view that anticipatory distractor suppression is implemented via a spatially specific increase in alpha-band activity (*Moorselaar and Slagter, 2020*). Previously observed modulations of pre-stimulus alpha-band activity may instead reflect attending away rather than suppression of distracting information, as also argued previously (*Foster and Awh, 2019*). Indeed, in our recent study using a similar visual search task, but in which the target or the distractor was repeatedly presented at the same location, across

four consecutive trials, the topographic distribution of preparatory alpha power was sensitive to the expected target location, but not to the expected distractor location (*van Moorselaar and Slagter, 2019*; see also *Noonan et al., 2016*). Together, these findings suggest that target facilitation and distractor suppression rely on distinct neural mechanisms, at least when space-based. Albeit speculative, distractor location learning may be implemented via an activity silent mechanism or synaptic efficiency (*Stokes, 2015*). An intuitively appealing idea, one that can also be easily reconciled with proposals that (spatial) expectations exert their influence only after initial processing of sensory input (*Alilović et al., 2019*; *Rao et al., 2012*), is that distractor learning changes plasticity in priority maps of space (*Ferrante et al., 2018*). Alternatively, it is possible that the spatial bias was evident in the neural signal, but our analysis, which focused on lateralized alpha power, was simply not sensitive enough to detect such a bias. Note, however, that in our previous work we also found no evidence for an anticipatory bias using either a multivariate decoding or a forward encoding approach, although that study provided considerably less time for distractor location predictions to develop (*van Moorselaar and Slagter, 2019*). Nonetheless, in line with a large body of work relating changes in lateralized pre-stimulus alpha-band activity to spatial attention (*Foxe and Snyder, 2011*; *Jensen and Mazaheri, 2010*), in that study, the expected target location was represented in the pattern of pre-stimulus alpha-band activity. Future work in humans and animals is thus necessary to determine whether changes in synaptic plasticity indeed underlie distractor location learning, or alternatively whether spatial tuning can be observed in the absence of lateralized alpha modulations with more sensitive methods, such as intracranial EEG.

A striking aspect of the present findings was that we did find evidence for distractor-specific feature templates based on the pattern of pre-stimulus brain activity when the spatial frequency of the distractor was predictable. The existence of so-called templates for rejection is a long-standing topic of debate in the domain of working memory research (*Arita et al., 2012*; *Beck and Hollingworth, 2015*). While behaviorally, there is some evidence in support of an advance distractor-specific inhibitory template (e.g. *Carlisle and Nitka, 2018*; *Woodman and Luck, 2007*), previous fMRI studies did not provide support for the notion of a distinct distractor template (*de Vries et al., 2019*; *Reeder et al., 2018*). Counter to these previous studies, which cued the distractor feature on a trial-by-trial basis, hence promoting the use of visual working memory, here the distractor 'template' could be formed on the basis of predictions derived from regularities in the search context. That we could decode the predictable distractor feature under these conditions thus suggests that a distinct distractor template may only arise as a function of learning from repeated encounters with the to-be-ignored feature. The development of this template may rely on implicit learning mechanisms that bypass working memory altogether (*Moorselaar and Slagter, 2020*) and hence should not be taken as evidence for the existence of templates for rejection in working memory. Nevertheless, they do convincingly show that distractor feature expectations, like target feature expectations, induce pre-stimulus sensory templates.

Note, however, that decoding is uninformative about the underlying neural representation (*Naselaris and Kay, 2015*; *Moorselaar and Slagter, 2020*). The here observed anticipatory distractor decoding could be driven by selective changes in the activity of neural populations representing the predicted distractor feature, selective changes in the activity of neural populations representing non-distractor or target (i.e. the other two spatial frequencies) features, or both. Indeed, it is well known that the visual system can strategically boost features shifted away from the distractor and/or target to increase discriminability between the target and the likely distractors (*Geng and Witkowski, 2019*; *Navalpakkam and Itti, 2007*). To determine whether the neural pattern underlying distractor decoding was distinct from the pattern driving target decoding, we also examined whether a model trained on distractor spatial frequencies generalized to target spatial frequencies. Strikingly, we found that while we could decode the anticipated spatial frequency of the target above chance in target-only trials prior to search display onset when the model was trained on target spatial frequencies, we could not decode the same anticipated spatial frequency of the target when the model was trained on distractor spatial frequencies. What's more, after search display onset there was reliable below-chance decoding, which arguably arose because after search display onset, attention was shifted to the target spatial frequency during training on the distractor spatial frequency, resulting in negative decoding when the classifier was tested on the target-only trials and the target now possessed the distractor spatial frequency. If distractor and target templates were opposite of one another, however, because observers strategically shifted their attention away from the anticipated

distractor spatial frequency towards the remaining potential target spatial frequencies, one would have expected to also observe reliable below chance decoding in the anticipatory period. By contrast, pre-stimulus target decoding performance of the distractor-trained model did not differ between the conditions in which the target spatial frequency could be predicted in advance vs. could not be predicted in advance, nor was there below chance decoding. Although these are exploratory null findings and therefore should be interpreted cautiously, they nevertheless argue against the idea that attention was strategically shifted away toward one (or both) of the remaining (and possible target) spatial frequencies, but rather support the notion of a distinct distractor template. Note also that it is unlikely that the observed anticipatory distractor feature decoding simply reflects inter-trial priming as in this scenario, one would not expect a difference in the neural activity patterns driving distractor decoding versus target decoding, as was observed here. Together, these decoding results indicate that target and distractor feature expectations are implemented in distinct neural codes. Future M/EEG studies that apply forward encoding modeling are necessary to more precisely examine the nature of the changes in feature tuning that underlie feature-based distractor suppression.

At the behavioral level, distractor interference was greatly reduced when distractors occurred at the high-probability distractor location. Yet, this benefit was not associated with active anticipatory spatial suppression as implemented by alpha-band activity, as discussed above, nor did we observe enhanced post-distractor suppression as indicated by the amplitude of the Pd ERP component (*Feldmann-Wüstefeld and Vogel, 2019*; *Hickey et al., 2009*). That is, replicating our previous study (*van Moorselaar and Slagter, 2019*), across conditions the distractor-evoked Pd was strongly reduced when distractors appeared at their expected location, as if the brain had learned that stimuli at those locations could be safely ignored, even though they still evoked a reliable N2pc. While a reduction of the Pd appears consistent with a dampened response to expected sensory input and thus less need for suppression, many studies have actually shown that expected distractors, either at the spatial (*Wang et al., 2019*) or the feature level (*Burra and Kerzel, 2013*; *Jannati et al., 2013*), elicit a Pd instead of an N2pc. Following the idea that the Pd signals a suppressive mechanism, which is stronger in contexts that require strong suppression (*Hickey et al., 2009*), these findings are typically interpreted to reflect the suppression of salient distractors before they capture attention (*Gaspelin and Luck, 2018b*). To date, it is unclear why in certain contexts the Pd increases, whereas in others, as in the current study, its amplitudes decreases as a function of expectations (*Heuer and Schubö, 2020*). Differences in distractor salience or in the presence of non-singleton distractors may explain this discrepancy in findings. A better understanding of the factors that modulate distractor learning and suppression is critical for advancing our understanding of learned distractor suppression and should thus be the focus of future work.

Our ERP findings also corroborate the notion that the N2pc and the Pd reflect different attentional mechanisms, as these components were differentially sensitive to feature and spatial expectations. Although it was initially thought that the N2pc reflects a covert shift of attention to a peripheral stimulus (*Eimer, 1996*; *Woodman and Luck, 1999*), current evidence suggests that the N2pc indexes object individuation (*Mazza and Caramazza, 2015*), the formation of an object representation that is segregated from the background and other items in the display (*Kahneman et al., 1992*). This idea dovetails with our finding that N2pc amplitude was modulated by expectations at the feature level, but not by spatial expectations. The specific perceptual process reflected by the N2pc, however, continues to be the topic of active debate. In one perspective, which is easily reconciled with the idea of object individuation, the N2pc reflects attentional engagement, which according to *Zivony et al., 2018* reflects the deployment of higher level processes that enable identification and binding of stimulus features, and, if necessary, consolidation into working memory. Consistent with this perspective, here we show that when the spatial frequency of both the target and the distractor were predictable, and distractors and targets could thus easily be dissociated, the distractor-evoked N2pc was smaller in amplitude, whereas the target-evoked N2pc increased in amplitude. By contrast, as outlined above, the Pd was only sensitive to spatial expectations in that its amplitude was greatly reduced to distractors at the high-probability distractor location. This demonstrates that the Pd, which is often used to index stimulus-driven distractor inhibition (*Gaspelin and Luck, 2018b*; *Sawaki et al., 2012*), is specifically sensitive to learned expectations at the spatial level.

Although there is growing evidence that regularities at both the spatial and the feature level shape attentional orienting, it is unclear to what extent, if at all (*Wang and Theeuwes, 2018b*), feature-based expectations can shape attentional selection above and beyond location-based suppression (*Stilwell et al., 2019*). At the behavioral level, we found that distractor suppression at the high-probability distractor location, but not target suppression at the high-probability distractor location, was affected by stimulus feature regularities. This may suggest that while spatial distractor expectations may induce feature-blind 'proactive' suppression (which uniformly suppresses both distractor and target processing at the expected distractor location), subsequent inhibition of stimuli at the likely distractor location is dependent on expectations about what a distractor looks like at the feature level. This makes sense from an ecological perspective: one would not want to simply inhibit anything occurring at a particular location in space. While the interaction between spatial and feature expectations in the distractor tuned behavioral analyses was significant in both Experiments 1 and 2, after priming effects were removed, it only remained significant in Experiment 2. This may suggest that the interaction also to some extent reflects an interaction between priming and distractor feature expectations, but could also reflect the fact that Experiment 2 allowed for the development of more robust feature and location expectations. That is, counter to Experiment 2, in Experiment 1, conditions not only switched every block (and consequently also the predictable distractor and/or target features), but participants also only completed six blocks per condition. In Experiment 2, distractor location and feature learning could develop over a much longer time scale (18 blocks), and participants completed 36 blocks per condition. Consequently, Experiment 1 may have been less sensitive to detecting the interaction between feature and spatial expectations above and beyond intertrial priming. The fact that the interaction remains reliable in Experiment 2, however, suggests that these different types of expectations can interact above and beyond intertrial biases.

It may seem surprising at first that none of our EEG measures showed an interaction between spatial and feature expectations, while in the corresponding behavioral data, we found that distractor interference was most reduced when both the location and the spatial frequency of the distractor were predictable in advance. Yet, this may be explained by the fact that these EEG measures are specifically tuned to capturing modulations in either spatial processing (i.e. spatially lateralized measures (alpha asymmetry and Pd)) or feature representation (spatial frequency decoding). As such, they may be less sensitive to revealing neural mechanisms that integrate spatial and feature distractor foreknowledge. Indeed, our EEG data did show robust main effects of spatial and feature expectations.

To summarize, our findings add to a rapidly growing body of work showing that the ability to inhibit distracting information critically relies on expectations grounded in statistical regularities in the environment. They extend this work by showing that feature-based distractor expectations, but not location-based distractor expectations, modulate pre-stimulus representations, and in a distinct manner from expectations about upcoming target features, indicative of a distractor template. These findings have important implications for cognitive (neuroscience) theories of selective attention and predictive processing.

## Materials and methods

### Participants

A planned number of 18 participants (average (*M*) age = 23 years, range 19–27; seven men) participated in Experiment 1, and, after replacement of three participants (because preprocessing resulted in exclusion of too many trials; details see below), a planned number of 24 participants (*M = 22 years old*, age range = 19–26; five men) participated in Experiment 2, in exchange for course credit or monetary compensation (10 euros/hr). Participants reported normal or corrected-to-normal vision and provided written informed consent prior to participation. The ethical committee of the Department of Psychology of the University of Amsterdam approved the study (2018-BC-9051), which was conformed to the Declaration of Helsinki.

## Task, stimuli, and procedure

### Experiment 1

The paradigm was adapted from our initial study (*van Moorselaar and Slagter, 2019*). As can be seen in *Figure 1*, trials started with a fixation display (500 ms) that only contained a central black dot with a white rim (radius 0.1°). A placeholder display was subsequently presented for 250 ms, after which a search display was shown for 200 ms or until response. The next trial started after response collection, or in case of no response, 1000 ms after search display onset.

The placeholder display contained six black rimmed circles (radius 1.5°), all placed on an imaginary circle (radius 4.6°) centered on fixation. At search display onset two Gabor patches (contrast = 1) were presented within the centers of two selected placeholders. One Gabor, the target, was tilted left or right (45° and 135°), while the other Gabor, the distractor, was vertically or horizontally oriented. Although the target and the distractor could appear at all six stimulus positions, the distractor appeared on one particular location more frequently (70%; location counterbalanced across participants) than on any of the other locations. In each trial, the target location was selected at random, once the distractor location was determined.

Crucially, the target and the distractor could either have a high (sf = 0.05) or a low (sf = 0.02) spatial frequency. In the DpTp identical condition, the distractor and the target had the same spatial frequency (e.g. both high; counterbalanced across blocks). In the DpTp unique condition, the spatial frequency of the target and the distractor differed, but they were also fixed within a block (counterbalanced across blocks). In the DvTv unique condition, the target and distractor had different spatial frequencies that varied from trial to trial (counterbalanced across trials). Thus, in both the DpTp identical and DpTp unique condition, the target and distracter spatial frequency were predictable in a given block, but not in the DvTv unique condition. Yet, only the DpTp unique condition allowed for both distractor and target-specific predictions at the feature level.

Participants first completed 50 practice trials in which the target and the distractor had the same spatial frequency, and then 18 experimental blocks (six blocks for each condition) of 50 trials each, with condition order (DpTp identical, DpTp unique, DvTv unique, ….., etc) counterbalanced over participants. They were instructed to respond as fast as possible, while trying to keep the number of errors to a minimum. At the end of each block, participants received feedback about response time and accuracy and were encouraged to take a break. At start of each block, participants were informed about the upcoming condition.

### Experiment 2

The task in Experiment 2 was similar to Experiment 1, but we included important changes (see *Figure 1*). In the DpTp identical condition and in the DpTp unique condition, the spatial frequencies of the target and distractor were again fixed and hence predictable within a block of trials, with the main difference between conditions that in the former condition the target and distractor shared the same spatial frequency, whereas in the latter their spatial frequencies differed and could guide attentional selection. In the DpTv unique condition, the distractor and target also had unique spatial frequencies, but only the spatial frequency of the distractor was fixed and hence predictable, whereas the target spatial frequency was selected at random. Targets and distractors could have one of three spatial frequencies (low sf = 0.02; medium sf = 0.04; high sf = 0.08). To prevent a systematic bias in the decoding analysis, in the DpTp and DpTv unique conditions, spatial frequencies were selected such that across all blocks, a specific spatial frequency serving as a target or a distractor was equally often coupled to the remaining spatial frequencies serving as a distractor or a target, respectively. Also, in Experiment 2, to be able to focus on lateralized components in the EEG signal, the high-probability distractor location (i.e. 70% of distractor present trials) could only be one of the lateralized locations below the horizontal midline. This location was fixed within a condition and shifted to the other side of the display after all trials of a given condition were completed, with the high-probability location and condition order counterbalanced across participants (e.g. 18 consecutive blocks DpTp identical, 18 consecutive blocks DpTv unique, 18 consecutive blocks DpTp unique). Finally, each block contained six randomly intermixed distractor absent trials in which the target was presented in isolation at each location once.

Participants came to the lab twice. In each session, they completed 54 blocks of the task of 56 trials each, wherein condition changed every 18 blocks. Simultaneously, we measured their brain

activity with EEG and their eye movements using eye tracking and EOG. In the first session, participants first completed a series of 56 practice trials of the DpTp identical condition. Manual responses were collected via two purpose-built response buttons, which were positioned at the end of the armrests of the participant's chair. In the search task, the length of the fixation display preceding the placeholder display was randomly jittered between 650 ms and 1000 ms.

In both experiments, a Windows 7 PC running OpenSesame v3 (*Mathôt et al., 2012*) using PsychoPy (*Peirce, 2008*) functionality generated the stimuli on an ASUS VG236 120 Hz monitor with a grey background, at ~80 cm viewing distance. Participants sat in a dimly lit room.

## Analyses

### Software

All preprocessing steps and most analyses were performed in a Python environment (Python Software Foundation, https://www.python.org/). Preprocessing and subsequent analyses scripts are publicly available and can be downloaded at https://github.com/dvanmoorselaar/DvM (*van Moorselaar, 2020*). These custom written analysis scripts are largely based on MNE functionalities (*Gramfort et al., 2014*). Repeated measures ANOVA and planned comparisons with paired t-tests were done using JASP software (*JASP-TEAM, 2018*). For all non-significant results, we also report the Bayes factor to quantify the evidence in favor of the null hypothesis ($BF_{01}$).

## Behavioral preprocessing

As preregistered, behavioral analyses were limited to RT data of correct trials only. RTs were filtered in a two-step trimming procedure: trials with RTs shorter than 200 ms were excluded, after which data were trimmed on the basis of a cutoff value of 2.5 SD from the mean per participant. Exclusion of incorrect responses (13.6% in Experiment 1; 7.9% in Experiment 2) and data trimming (2.0% in Experiment 1; 2.6% in Experiment 2) resulted in an overall loss of 15.6% of the data in Experiment 1% and 10.5% of the data in Experiment 2. Analyses were either yoked to the distractor location, in which case trials with targets at the high-probability distractor location were excluded, or to the target location, in which case trials with distractors at the high-probability location were excluded. This control made sure reported effects were not inflated, but more specific to distractor and target suppression, respectively, at the high-probability distractor location.

## EEG recording and preprocessing

EEG data were recorded at a sampling rate of 512 Hz using a 64-electrode cap with electrodes placed according to the 10–10 system (Biosemi ActiveTwo system; biosemi.com) and from two earlobes (used as offline reference). Vertical (VEOG) and horizontal EOG (HEOG) were recorded via external electrodes placed ~2 cm above and below the eye, and ~1 cm lateral to the external canthi, respectively. All preprocessing steps were performed on the EEG data of the two sessions separately.

EEG data were re-referenced off-line to the average of the left and the right earlobe, and subsequently high-pass filtered using a zero-phase 'firwin' filter at 0.1 Hz to remove slow drifts. Epochs were then created from −750 ms to 550 ms surrounding search display onset, extended by 500 ms at the start and end of the epoch to control for filter artifacts during preprocessing and time-frequency analysis. The resulting epochs were baseline normalized using the whole epoch as a baseline to aid detection of noisy electrodes based on visual inspection. Prior to trial rejection malfunctioning electrodes ($M$ = 1.2, range = 0–5) were temporarily removed based on visual inspection. Epochs contaminated by EMG noise were identified using an adapted version of an automatic trial-rejection procedure as implemented in the Fieldtrip toolbox (*Oostenveld et al., 2011*). To specifically identify epochs contaminated by muscle activity, we used a 110–140 Hz band-pass filter, and allowed for variable z-score cut-offs per subjects based on within-subject variance of z-scores (*de Vries et al., 2017*) resulting in the identification of on average 10.2% of epochs (range 2.9–19.7%) as potentially containing artifacts. Marked epochs were visually inspected, resulting in an average rejection of 8.1% of epochs (range 1.4–17.0%). Next, ICA as implemented in the MNE 'extended-infomax' method was performed on non-epoched 1 Hz high-pass filtered data to identify and remove eye-blink components from the epoched data. Following ICA, malfunctioning electrodes identified earlier were interpolated using spherical splines (*Perrin et al., 1989*), after which the EEG data of the

separate sessions was concatenated. Finally, we detected sudden jumps in the HEOG using a step algorithm, with a window length of 200 ms, a step size of 10 ms, and with a threshold of 20 $\mu V$. The detected jumps using these settings were again visually inspected, which resulted in a rejection of 4.0% (range 0.4–11.4%) of the remaining epochs after artifact rejection (Counter to the preregistration we did not use the EyeTribe data to exert control over which epochs to include in the analysis as in a large subset of subject's calibration required too much time, or did not work at all. We therefore relied on a two-step process where we visually inspected epochs selected by a step algorithm (for details see main text)).

## Time-frequency analysis

To isolate frequency-specific activity across time, we used Morlet wavelet convolution to decompose EEG time series into their time-frequency representation for frequencies ranging from 1 to 40 Hz in 25 logarithmically spaced steps. To create complex Morlet wavelets, for each frequency a sine wave ($e^{i2\pi ft}$, where $i$ is the complex operator, $f$ is frequency, and $t$ is time) was multiplied by a Gaussian ($e^{-t^2/2s^2}$, where s is the width of the Gaussian). To keep a good trade-off between temporal and frequency precision the Gaussian width was set as $s = \delta/(2\pi f)$, where $\delta$. represents the number of wavelet cycles, in 25 logarithmically spaced steps between 3 and 12. Frequency-domain convolution was then applied by multiplying the Fast Fourier Transform (FFT) of the EEG data and Morlet wavelets. The resulting signal was converted back to the time domain using the inverse FFT. Time-specific frequency power, which was downsampled by a factor of 4, was defined as the squared magnitude of the complex signal resulting from the convolution.

To investigate whether the high-probability distractor location resulted in changes in the topographic distribution of alpha-power for each condition we calculated a lateralization index, in which raw power in each hemisphere (i.e. contralateral and ipsilateral relative to the high-probability distractor location) is expressed relative to the total power at both sites. This number is positive when contralateral power is larger than ipsilateral power and negative when the inverse is the case. This procedure yields a metric that shows interpretable dynamics over time and frequency and has the benefit of not having to select a pre-stimulus interval (i.e. the interval that we are interested in) for baseline normalization. Statistical analyses were limited to electrode pairs PO7/8 and O1/2, which were selected on the basis of visual inspection of the topographic distribution of averaged alpha-power (8–12 Hz) across the anticipatory time window (−750 to 0 ms).

## Decoding analysis

To test whether feature-specific predictions modulated the neural representation of spatial frequency, we applied multivariate pattern analysis (MVPA), using linear discriminant analysis (*Pedregosa et al., 2011*) in combination with a 10-fold cross validation scheme, with all 64 channels as features and target or distractor spatial frequency as classes. As we were specifically interested in anticipatory effects, which we expected to be most pronounced, if at all present, immediately preceding search display onset, preprocessed data was baseline corrected using a −750 ms to −550 ms window and subsequently down-sampled to 128 Hz to decrease the computational time of the MVPA analysis. EEG data was randomly split into 10 equally sized subsets, while ensuring that each class (i.e. low, medium, high spatial frequency) was selected equally often in each condition so that training was not biased toward a specific class. Next, a leave-one-out procedure was used on the 10 folds, such that classifier was trained on nine folds and tested on the remaining fold until each fold was tested once. Classifier performance was then averaged over folds. As a measure of classifier performance we used the Area Under the Curve (AUC), where a value of 0.5 is considered chance classification, and which is considered a sensitive, nonparametric and criterion-free measure of classification (*Hand and Till, 2001*). This analysis was conducted for every time point, showing how classifier performance changed over time.

## ERP analysis

For the ERP analysis we followed the same procedure as in our previous work (*van Moorselaar and Slagter, 2019*). To isolate distractor-specific ERP components, separate waveforms were computed for ipsilateral and contralateral scalp regions for lateralized distractors presented below the horizontal midline with concurrent targets on the vertical midline using O1/O2, PO3/PO4 and PO7/PO8 as

electrodes of interest. Epochs were 30 Hz low-pass filtered and baseline corrected using a 200 ms window preceding placeholder onset. The N2pc window was selected by taking the most negative peak in the condition-averaged difference waveform (contralateral – ipsilateral) of both targets and distractors in a 100 ms to 400 ms window post search display onset extended by 40 ms on both sides of the peak. The same approach was used to determine the Pd window of interest, but now searching the positive peak and only using distractor tuned waveforms.

## Statistics

RTs, and averaged data across time windows of interest (in time-frequency, decoding and ERP) were analyzed with repeated measures ANOVAs. Reported p-values are Greenhouse-Geiser corrected in case of sphericity violations. In case of relevant significant effects, we followed up by planned comparisons with paired t-tests, and in case of insignificant findings we report the Bayes factor to evaluate the null hypothesis. Time windows with significant effects were identified using cluster-based permutation testing across time and across time and frequency with cluster correction (p=0.05 and 1024 iterations) using MNE functionality (*Gramfort et al., 2014*).

## Acknowledgements

This research was supported by a European Research Council (ERC) starting grant (679399) to HAS.

## Additional information

### Funding

| Funder | Grant reference number | Author |
|---|---|---|
| H2020 European Research Council | 679399 | Heleen A Slagter |

The funders had no role in study design, data collection and interpretation, or the decision to submit the work for publication.

### Author contributions

Dirk van Moorselaar, Conceptualization, Software, Formal analysis, Supervision, Investigation, Visualization, Methodology, Writing - original draft, Writing - review and editing; Eline Lampers, Elisa Cordesius, Conceptualization, Investigation, Methodology, Writing - review and editing; Heleen A Slagter, Conceptualization, Supervision, Funding acquisition, Writing - review and editing

### Author ORCIDs

Dirk van Moorselaar (iD) https://orcid.org/0000-0002-0491-1317
Heleen A Slagter (iD) http://orcid.org/0000-0002-4180-1483

### Ethics

Human subjects: The ethical committee of the Department of Psychology of the University of Amsterdam approved the study (2018-BC-9051), which was conformed to the Declaration of Helsinki.

### Decision letter and Author response

Decision letter https://doi.org/10.7554/eLife.61048.sa1
Author response https://doi.org/10.7554/eLife.61048.sa2

## Additional files

### Supplementary files

• Transparent reporting form

## Data availability

All data are publicly available on OSF (https://osf.io/7ek45/). Analysis scripts can be downloaded via GitHub.

The following dataset was generated:

| Author(s) | Year | Dataset title | Dataset URL | Database and Identifier |
|---|---|---|---|---|
| van Moorselaar D, Lampers E, Cordesius E, Slagter HA | 2020 | Manipulating target distractor similarity | https://osf.io/7ek45/ | Open Science Framework, 7ek45 |

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
