## [Decision Letter]

**Acceptance summary:**

The two reviewers and I all found your work dissociating spatial and feature-based effects of distractor suppression innovative and exciting. I think your work will make a nice contribution to the growing literature on distractor processing.

**Decision letter after peer review:**

Thank you for submitting your article "Neural mechanisms underlying expectation-dependent inhibition of distracting information" for consideration by *eLife*. Your article has been reviewed by three peer reviewers, one of whom is a member of our Board of Reviewing Editors, and the evaluation has been overseen by Chris Baker as the Senior Editor. The following individuals involved in review of your submission have agreed to reveal their identity: Keisuke Fukuda (Reviewer #2); Jason Samaha (Reviewer #3).

The reviewers have discussed the reviews with one another and the Reviewing Editor has drafted this decision to help you prepare a revised submission.

Summary:

van Moorselaar et al. use EEG and behavior to examine preparatory and stimulus-evoked signatures of distractor suppression as a function of spatial and feature-based expectancies.

Although distractor suppression has been studied through behavioral methods for some time, there have been very few studies using non-invasive methods in humans to understand how the brain implements suppression of task-irrelevant information. The authors use a paradigm in which the spatial location and the features of the distractors are predictable and use EEG to measure how that information is used to suppress processing and reduce distractor interference.

The reviewers commented that the research question is timely and builds on a literature of how expectations shape pre-stimulus brain activity, but now applied to the topic of space and feature-based suppression of distractors.

Essential revisions:

The reviewers raise a number of concerns that must be adequately addressed before the paper can be accepted. Some of the required revisions may require extensive re-analyses of the data.

1) It was quite interesting that the authors found feature-based expectations about the distractor to be present in the prestimulus period but not space-based expectations. However, this conclusion is predicated of "fair" comparisons of the two sources of information. However, the effects are based on different analyses. Is it possible that the space-based effects appear weaker because the multivariate analyses used for feature-based decoding are more sensitive than the alpha lateralization used for analysis of space-based effects? To address this, the authors should apply the same decoding approach to both voltage and alpha power to see if a prestimulus spatial representation exists.

Another reason why lateralized alpha might be less sensitive is that it is too broadly defined to capture spatially specific suppression effects at the distractor location. The paradigm uses six possible target-distractor locations and only one location has a high probability of containing the distractor; the target location is random. Wouldn't lateralized alpha (on a single trial) contain information about not only the expected distractor but also other locations that could contain the target? In theory suppression should be applied more narrowly than alpha lateralization over an entire visual field. If more than one location was included in the analysis, then it needs to be clearer how stimulus laterality was determined. Were midline distractor locations excluded from the analyses? Were the two stimulus positions on the left or right grouped together to define "left" and "right" stimuli? If so, both of these choices would potentially reduce power by (1) throwing out trials, and (2) averaging over potentially distinct topographical representations.

(2) Several questions were raised regarding the timing of the prestimulus analyses. One worry was that the late prestimulus decoding is due to smearing of the stimulus-evoked response. If the average of the entire epoch was subtracted from each trial, this could result in the biased removal of signal associated with stimulus types that elicit a larger response. Additionally, is it possible that the baselining procedure "subtracted out" sustained expectation signals that were present throughout the pre-stimulus interval, causing them to appear absent?

There was also a more general point raised about the possibility that the z-transform approach used to analyze alpha may not be optimal since it only makes sense for a roughly normal signal. Is the lateralized signal normally distributed? log10 transform of raw power has been suggested as an optimal transform of alpha power and would help normalize the signal.

3) Reviewers thought the interpretation of the behavioral interactions, particularly in regard to differences in suppression effects on targets and distractors, should be clarified. Target processing in the high probability distractor location was slowed uniformly without modulation by feature predictability. This suggests a fixed level of suppression tied to the location. However, when distractors are in the high probability location, suppression is modulated by feature, suggesting an interaction between spatial probability and features. This inconsistency between targets and distractors was somewhat perplexing and was not fully integrated with the EEG results. This should be more fully explored or explained. The authors should also comment on differences in the pattern of interactions between Experiment 1 and Experiment 2 (including interpretation of results in Experiment 1 when priming effects are removed; and possible ceiling effects in the DpTp conditions).

4) The decoding data provided an interesting finding of unique information for targets and distractors. This suggests that the target template and distractor template are implemented with distinct codes but this interpretation depends on interpretation of the general pattern of results. However, the specific decoding accuracy of each condition should be more fully discussed. Specifically, the finding that some conditions involved apparent at chance decoding (Figure 4) or below chance decoding (Figure 5) are puzzling and should inform interpretation of target and distractor templates and their relationship (e.g., none or opposite) to each other.

5) The fact that N2pc did not differ based on whether the distractor was in a high or low probability location (suggesting similar attentional processing) but that the Pd was smaller when the distractor was in the high probability location suggests that these two components are sensitive to different information but this is not fleshed out given that it runs contrary to some interpretations of these components in the literature.

---

## [Author Response]

Essential revisions:The reviewers raise a number of concerns that must be adequately addressed before the paper can be accepted. Some of the required revisions may require extensive re-analyses of the data.1) It was quite interesting that the authors found feature-based expectations about the distractor to be present in the prestimulus period but not space-based expectations. However, this conclusion is predicated of "fair" comparisons of the two sources of information. However, the effects are based on different analyses. Is it possible that the space-based effects appear weaker because the multivariate analyses used for feature-based decoding are more sensitive than the alpha lateralization used for analysis of space-based effects? To address this, the authors should apply the same decoding approach to both voltage and alpha power to see if a prestimulus spatial representation exists.

This is a valid point and we agree that a potential explanation for the absence of anticipatory spatial tuning in our study could be that alpha lateralization was not sensitive enough to detect any space-based effects, although we think this is not likely (as explained below). Unfortunately, we could not apply a decoding approach to examine pre-stimulus spatial representations as, even though the high probability distractor location could be one of the lateralized locations below the horizontal midline, this location was fixed within all blocks of a given condition (i.e., the high probability distractor location only changed when a new condition started; see subsection “Experiment 2”). Consequently, decoding of spatial location would render it unclear whether any above chance decoding is driven by a spatial bias and/or condition (as these two were always coupled).

Although we cannot fully exclude the possibility that the alpha lateralization used for the analysis of space-based effects may not have been sensitive enough to detect anticipatory spatial alpha-band tuning, we believe this is unlikely for several reasons. First, behaviorally we observed strong suppression at the high probability distractor location. Second, lateralized pre-stimulus alpha power is a robust effect widely reported in many previous EEG studies of spatial attention. Finally, in a previous study in which we applied multivariate decoding and inverted encoding modeling, we also did not observe any changes in spatial tuning to the expected distractor location (van Moorselaar and Slagter, 2019). Here we extend this and another recent EEG study (Noonan et al., 2016) that also failed to observe any modulation of lateralized pre-stimulus alpha activity as a function of distractor learning by showing that even when predictions can only be developed at the spatial level (not the feature level), there is no evidence that alpha oscillations contributed to the observed behavioral suppression. Note that these studies did show a role for anticipatory alpha in implementing target-based expectations, arguing against a general lack of sensitivity of this measure.

Nevertheless, we agree with the reviewers that the absence of this effect does not necessarily mean there was no active spatial tuning as our analysis may simply not have been sensitive enough. We have rephrased our Discussion accordingly.

“Alternatively, it is possible that the spatial bias was evident in the neural signal, but our analysis, which focused on lateralized alpha power, was simply not sensitive enough to detect such a bias. […] Future work in humans and animals is necessary to determine whether changes in synaptic plasticity indeed underlie distractor location learning, or alternatively whether spatial tuning can be observed in the absence of lateralized alpha modulations with more sensitive methods, such as intracranial EEG.”

Another reason why lateralized alpha might be less sensitive is that it is too broadly defined to capture spatially specific suppression effects at the distractor location. The paradigm uses six possible target-distractor locations and only one location has a high probability of containing the distractor; the target location is random. Wouldn't lateralized alpha (on a single trial) contain information about not only the expected distractor but also other locations that could contain the target? In theory suppression should be applied more narrowly than alpha lateralization over an entire visual field. If more than one location was included in the analysis, then it needs to be clearer how stimulus laterality was determined. Were midline distractor locations excluded from the analyses? Were the two stimulus positions on the left or right grouped together to define "left" and "right" stimuli? If so, both of these choices would potentially reduce power by (1) throwing out trials, and (2) averaging over potentially distinct topographical representations.

In the analysis of alpha-band activity we selectively focused on the anticipatory period, where by design observers could critically only have developed expectations about the upcoming distractor location, which either appeared with a higher probability at the bottom left or bottom right position in the search display. How this was used to determine alpha lateralization is now clarified in the subsection “Time-frequency analyses: Pre-stimulus alpha-band activity does not increase contralateral to the high probability distractor location”. Consequently, any robust lateralization cannot be attributed to attending towards the expected target location as may have been the case in previous studies examining distractor suppression with binary displays, where foreknowledge about the distractor location inevitably also reveals the upcoming target location, and vice versa. By contrast, here in distractor present trials, the target location was selected at random with the restriction that it did not match the distractor location. Nevertheless, even if the anticipatory activity contained information about the other stimulus locations, we would not expect this to affect *lateralized* alpha activity, as these locations were both within the same and the other hemifield as the likely distractor location. Moreover, as we specifically focused on the anticipatory period, where by design spatial expectations were exclusive to either the bottom right or bottom left of the display, all experimental trials could be included in the analysis. Therefore, we believe it is unlikely that the observed null findings can be attributed to reduced power or a smeared average over distinct topographical representations.

2) Several questions were raised regarding the timing of the prestimulus analyses. One worry was that the late prestimulus decoding is due to smearing of the stimulus-evoked response. If the average of the entire epoch was subtracted from each trial, this could result in the biased removal of signal associated with stimulus types that elicit a larger response. Additionally, is it possible that the baselining procedure "subtracted out" sustained expectation signals that were present throughout the pre-stimulus interval, causing them to appear absent?

These are valid concerns. We were specifically interested in endogenously generated anticipatory space and feature-based tuning. Therefore, it is not immediately clear what window to select as a pre-stimulus baseline, as that same window, as also pointed out by the reviewers, might contain effects of interest as well. Note, however, that this is only a potential concern for the decoding analysis, as in the time-frequency analysis we adopted a method (see more details in our response to the concern about the Z-transform approach) that does not require taking a pre-stimulus baseline. By contrast, in the decoding analysis, we used a pre-stimulus baseline (-750 to -550ms; see subsection “Decoding analysis”), not the average of the entire epoch as stated above. The latter was only done to aid visual inspection (as now clarified in the subsection “EEG recording and preprocessing”). It is indeed possible that the adopted baselining procedure subtracted out anticipatory signals that were present in the pre-stimulus baseline window. To explore this, in two control analysis we repeated the same decoding analysis either without a pre-stimulus baseline and with a condition-specific baseline (i.e., collapsed across spatial frequency). As both these analyses (see Author response image 1) yielded virtually identical results we believe it unlikely that our baselining procedure obscured any anticipatory effects. We have added this information to the revised manuscript subsection (“Decoding: Target and distractor-based feature expectations induce pre-stimulus sensory templates”) (and see Author response image 1).

There is also still the risk that the high-pass filter (0.1Hz) applied during preprocessing artificially shifted the stimulus-evoked response (van Driel, Olivers and Fahrenfort, 2019), resulting in a potential smearing of the stimulus-evoked response into the late pre-stimulus decoding window. We also believe this to be unlikely. Anticipatory target decoding was absent in the DpTv unique condition, the one condition where target expectations could not develop, despite robust reactive target decoding in this condition. If smearing of the stimulus-evoked response contributed to our decoding findings, we should also see it in this condition, but we don’t.

“Critically, anticipatory target decoding was absent in the DpTv unique condition, the one condition where target expectations could not develop, arguing against the possibility that the observed anticipatory decoding can be explained by an artificial shift of the stimulus-evoked response (van Driel, Olivers and Fahrenfort, 2019). Also note that anticipatory above chance decoding was not sustained throughout the entire pre-stimulus period, but only emerged close to search display onset, raising the possibility that sustained anticipatory expectation signals were subtracted out by the baselining procedure (see Materials and methods). Control analyses, however, using either a condition-specific pre-stimulus baseline (collapsed across spatial frequency) or no baseline at all, yielded identical results indicating that the observed feature specificity in the EEG signal only emerged in close temporal proximity to search display onset.”

**Author response image 1. respfig1:** Multivariate decoding of the target and distractor spatial frequency across conditions using broad-band EEG of all 64 electrodes. (**A**) AUC scores of target (left) and distractor (right) decoding across time without a pre-stimulus baseline. (**B**) AUC scores of target (left) and distractor (right) decoding across time with a condition specific pre-stimulus baseline. Colored bars on the x-axis (blue; red; green) indicate clusters where conditions differ significantly from chance after cluster correction (p <0.05). Black lines indicate clusters with a significant difference between DpTv unique and DpTp unique conditions after cluster correction (p <0.05).

There was also a more general point raised about the possibility that the z-transform approach used to analyze alpha may not be optimal since it only makes sense for a roughly normal signal. Is the lateralized signal normally distributed? log10 transform of raw power has been suggested as an optimal transform of alpha power and would help normalize the signal.

This is another relevant point. As explained in the Manuscript and in response to the previous comment, we used the Z-transform approach, because this method has the benefit of not having to select a pre-stimulus baseline window. A pre-stimulus baseline can be problematic if the effect of interests overlaps with the window used for baselining, as was also pointed out by the reviewers. For this reason, the Wang et al., 2019 study, the one study that observed anticipatory alpha-band tuning to expected distractor locations, developed the Z-transform method and we followed their methods. An alternative method to normalize the data without using a baseline window, however, is to calculate a lateralization index, in which power values at every ipsilateral electrode is subtracted from its contralateral electrode and then normalized by the total power at both electrodes. (i.e., contralateral plus ipsilateral). Since this method yielded virtually identical results, and as pointed out by the reviewers, the Z-transform approach may not have been optimal, we now report this analysis (see subsections “Time-frequency analyses: Pre-stimulus alpha-band activity does not increase contralateral to the high probability distractor location” and “Time-frequency analysis”). We also report that the pattern of results did not change when we used Z-transform approach as reported in Wang et al., 2019.

3) Reviewers thought the interpretation of the behavioral interactions, particularly in regard to differences in suppression effects on targets and distractors, should be clarified. Target processing in the high probability distractor location was slowed uniformly without modulation by feature predictability. This suggests a fixed level of suppression tied to the location. However, when distractors are in the high probability location, suppression is modulated by feature, suggesting an interaction between spatial probability and features. This inconsistency between targets and distractors was somewhat perplexing and was not fully integrated with the EEG results. This should be more fully explored or explained. The authors should also comment on differences in the pattern of interactions between Experiment 1 and Experiment 2 (including interpretation of results in Experiment 1 when priming effects are removed; and possible ceiling effects in the DpTp conditions).

We thank the reviewers for these comments. We agree that the apparent differences in the distractor tuned and target tuned behavioral analyses should be discussed in more detail. The fact that at the behavioral level, we find that distractor suppression at the high probability distractor location, but not target suppression at the high probability distractor location, is affected by stimulus feature regularities, suggests that while spatial distractor expectations may induce feature-blind “proactive” suppression (which uniformly suppresses both distractor and target processing at the expected distractor location), subsequent inhibition of stimuli at the likely distractor location is dependent on expectations about what a distractor looks like at the feature level. In other words, the system appears to combine spatial and feature information about probable upcoming distractors to reduce distractor interference. This makes sense from an ecological perspective: one would not want to simply inhibit anything occurring at a particular location in space. Thus, while at first sight, the inconsistency between targets and distractors may seem somewhat perplexing, it very much makes sense that the effects from spatial distractor expectations are also integrated with distractor feature expectations.

Yet, as is pointed out by the reviewers, while the interaction between spatial and feature expectations in the distractor tuned behavioral analyses was significant in both Experiments 1 and 2, after priming effects were removed, it only remained significant in Experiment 2. This suggests that the interaction also to some extent reflects an interaction between priming and distractor feature expectations. At the same time, we would like to stress that counter to Experiment 2, in Experiment 1, conditions not only switched every block (and consequently also the predictable distractor and/or target features), but participants also only completed 6 blocks per condition. In Experiment 2, distractor location and feature learning could develop over a much longer time scale (18 blocks), and participants completed 36 blocks per condition. Consequently, Experiment 1 may have been less sensitive to detecting the interaction between feature and spatial expectations above and beyond intertrial priming. The fact that the interaction remains reliable in Experiment 2, however, suggests that these different types of expectations can interact above and beyond intertrial biases. In the revised version we now highlight these differences and discuss them already when we summarize the effect of Experiment 1:

“Specifically, target processing at the high probability distractor location was slowed uniformly across conditions (independent of feature expectations), while the effects of spatial expectations on distractor interference, although present in each condition, were most pronounced in the absence of any feature expectations. […] It should be noted however that the observed interaction between spatial and feature expectations in the distractor-tuned analysis disappeared when we controlled for intertrial spatial priming, an aspect that we will revisit in Experiment 2, which contained many more trials per condition, and hence permitted for the development of more robust spatial and feature expectations.”

and of Experiment 2:

“Together, these results replicate the finding of Experiment 1 that the visual system continues to be sensitive to feature information at a location that is suppressed in advance (Stilwell et al., 2019). […] This makes sense from an ecological perspective: one would not want to simply inhibit anything occurring at a particular location in space.”

This then leaves the question how these behavioral observations can be integrated with our EEG findings that did not provide evidence for an interaction between distractor location and feature expectations? The latter may be due to the fact that our EEG measures are specifically tuned to capturing modulations in either spatial processing (i.e., spatially lateralized measures (alpha asymmetry and Pd)) or feature representation (spatial frequency decoding). They may hence be less sensitive to revealing neural mechanisms that integrate spatial and feature distractor foreknowledge. Indeed, our EEG data did show robust main effects of spatial and feature expectations (e.g., for the Pd ERP component). (note that at the behavioral level, the main effects are robust across the two experiments). In the revised Discussion, we now discuss this:

“Although there is growing evidence that regularities at both the spatial and the feature level shape attentional orienting, it is unclear to what extent, if at all (Wang and Theeuwes, 2018b), feature-based expectations can shape attentional selection above and beyond location-based suppression (Stilwell et al., 2019). […] Indeed, our EEG data did show robust main effects of spatial and feature expectations.”

4) The decoding data provided an interesting finding of unique information for targets and distractors. This suggests that the target template and distractor template are implemented with distinct codes but this interpretation depends on interpretation of the general pattern of results. However, the specific decoding accuracy of each condition should be more fully discussed. Specifically, the finding that some conditions involved apparent at chance decoding (Figure 4) or below chance decoding (Figure 5) are puzzling and should inform interpretation of target and distractor templates and their relationship (e.g., none or opposite) to each other.

As already shortly pointed out in our response to comment 2, whereas the DpTp unique condition allowed for feature expectations at both the distractor and the target level, in the DpTv unique condition such expectations could only be formed at the distractor level, but importantly not at the target level. Hence, any model in the anticipatory period trained and/or tested on target features in the DpTv condition should result in at chance decoding, which is indeed what we observed in the regular decoding analysis (Figure 4) and in the cross-category decoding analysis (Figure 5). Critically, in the cross-category decoding results, we also find that a classifier trained on distractor spatial frequency in the DpTp unique condition cannot decode that same spatial frequency pre-search when it is an expected property of the target, even though the regular (within category) decoding analysis showed that both targets and distractors could independently be decoded pre-search. Moreover, pre-stimulus decoding in this condition did not differ from decoding accuracy in the condition in which that same spatial frequency was a non-predictable property of the target and pre-search decoding was thus at chance. The fact that we could not decode the spatial frequency when it was a predicted feature of the upcoming target based on a classifier that could decode that same spatial frequency pre-search when it was a predicted feature of the upcoming distractor suggests that the target and distractor template prior to search are not related to each other. Spatial frequency must in other words be differentially represented in the pattern of scalp EEG activity as a function of whether it is an expected feature of the target or of the distractor.

The question then becomes: in what way? On this, we can only speculate. In the Discussion, we speculated that if the templates were opposite of each other (because distractor feature expectations strategically shifted tuning away from the anticipated distractor spatial frequency towards the possible target spatial frequencies), one would have expected *below* chance decoding pre-search (or more evidence for the possible target spatial frequencies in the pattern of pre-search scalp EEG activity). Indeed, after search display onset we do observe below-chance decoding, which one would expect given that attention then was oriented towards another spatial frequency than during training (that of the target during distractor-based training). But this is not what we find in anticipation of search, indicating that target and distractor feature expectations are implemented in distinct neural codes, and that distractor feature expectations did not simply induce a strategy of attending more to the possible target features. Future M/EEG studies that apply forward encoding modeling are necessary to more precisely examine the nature of the changes in feature tuning that make up the distractor template. In the revised manuscript, we already touched on the above, but we now more explicitly talk about the at chance and below chance decoding observed in the individual conditions, per the reviewers’ suggestion above.

“Strikingly, we found that while we could decode the anticipated spatial frequency of the target above chance in target-only trials prior to search display onset when the model was trained on target spatial frequencies, we could not decode the same anticipated spatial frequency of the target when the model was trained on distractor spatial frequencies. […] Although this is an exploratory null finding and therefore should be interpreted cautiously, these findings nevertheless argue against the idea that attention was strategically shifted away towards one (or both) of the remaining (and possible target) spatial frequencies, but rather support the notion of a distinct distractor template.”

5) The fact that N2pc did not differ based on whether the distractor was in a high or low probability location (suggesting similar attentional processing) but that the Pd was smaller when the distractor was in the high probability location suggests that these two components are sensitive to different information but this is not fleshed out given that it runs contrary to some interpretations of these components in the literature.

We thank the reviewers for this suggestion. Indeed, our data suggest that the distractor-evoked N2pc and Pd are sensitive to different information. The distractor-evoked N2pc was not modulated by distractor location expectations, while the distractor-evoked Pd was: the Pd was much smaller to distractors at the high probability distractor location. The N2pc finding may be surprising given previous proposals that the N2pc reflects the deployment of spatial attention (e.g., Eimer, 1996; Woodman and Luck, 1999), but fits with the now dominant view that the N2pc does not reflect spatial attention shifting, but a process of item individuation: the formation of an object representation that is segregated from the background and the other items in the display (Kiss et al., 2008; Mazza and Camarazza, 2015). Indeed, we found that the amplitude of the N2pc, while insensitive to spatial expectations, was modulated by feature expectations: the distractor-evoked N2pc was smallest when the spatial frequency of both the target and distractor could be uniquely predicted in advance, while the target-evoked N2pc was largest in this condition. This latter finding may suggest that feature expectations modulated object individuation in opposite ways for predicted targets and distractors, which we now also briefly discuss in the revised manuscript.

In contrast, the Pd component was sensitive to distractor location probability in that its amplitude was greatly reduced to distractors at the high distraction location, but this effect was independent of feature expectations, suggesting that this component is mainly sensitive to spatial information. The Pd is typically used to index reactive, stimulus-driven spatial inhibition, but the fact that it virtually disappeared as a function of distractor location learning suggests that it is also sensitive to learned spatial expectations. In the revised manuscript we now discuss the here observed differences in sensitivity to spatial and feature expectations between the N2pc and Pd with respect to current interpretations of these components. We also discuss how our ERP findings relate to findings from previous ERP studies showing that the N2pc and Pd reflect different, independent processes.

“Our ERP findings also corroborate the notion that the N2pc and the Pd reflect different attentional mechanisms, as these components were differentially sensitive to feature and spatial expectations. […] This demonstrates that the Pd, which is often used to index stimulus-driven distractor inhibition (Gaspelin and Luck, 2018b; Sawaki, Geng and Luck, 2012), is specifically sensitive to learned expectations at the spatial level.”